# Gfi1b regulates the level of Wnt/β-catenin signaling in hematopoietic stem cells and megakaryocytes

Peiman Shooshtarizadeh[1], Anne Helness[1], Charles Vadnais[1], Nelleke Brouwer[1], Hugues Beauchemin[1], Riyan Chen[1], Halil Bagci[1,2], Frank J.T. Staal [3], Jean-François Coté [1,2,4,5] & Tarik Möröy[1,6,7]

Gfi1b is a transcriptional repressor expressed in hematopoietic stem cells (HSCs) and megakaryocytes (MKs). Gfi1b deficiency leads to expansion of both cell types and abrogates the ability of MKs to respond to integrin. Here we show that Gfi1b forms complexes with β-catenin, its co-factors Pontin52, CHD8, TLE3 and CtBP1 and regulates Wnt/β-catenin-dependent gene expression. In reporter assays, Gfi1b can activate TCF-dependent transcription and Wnt3a treatment enhances this activation. This requires interaction between Gfi1b and LSD1 and suggests that a tripartite β-catenin/Gfi1b/LSD1 complex exists, which regulates Wnt/β-catenin target genes. Consistently, numerous canonical Wnt/β-catenin target genes, co-occupied by Gfi1b, β-catenin and LSD1, have their expression deregulated in Gfi1b-deficient cells. When Gfi1b-deficient cells are treated with Wnt3a, their normal cellularity is restored and Gfi1b-deficient MKs regained their ability to spread on integrin substrates. This indicates that Gfi1b controls both the cellularity and functional integrity of HSCs and MKs by regulating Wnt/β-catenin signaling pathway.

[1] Institut de recherches cliniques de Montréal, Montreal H2W 1R7 QC, Canada. [2] Department of Anatomy and Cell Biology, McGill University, Montréal, QC H3A 0C7, Canada. [3] Department of Immunohematology and Bloodtransfusion, Leiden University Medical Center, Leiden 2333 ZA, The Netherlands. [4] Programmes de Biologie Moléculaire, Département de Médecine, Université de Montréal, Montréal, QC H3T 1J4, Canada. [5] Département de Biochimie, Université de Montréal, Montréal, QC H3C 3J7, Canada. [6] Département de microbiologie, infectiologie et immunologie, Université de Montréal, C.P. 6128, succ. Centre-ville Montreal H3C 3J7 QC, Canada. [7] Division of Experimental Medicine, McGill University, Montreal H4A 3J1 QC, Canada. These authors contributed equally: Peiman Shooshtarizadeh, Anne Helness, Charles Vadnais. Correspondence and requests for materials should be addressed to T.M. (email: Tarik.Moroy@ircm.qc.ca)

*G*rowth factor independence 1b (Gfi1b) and its paralogue Gfi1 are transcription factors that are expressed in a complementary and partially overlapping manner in hematopoietic stem cells (HSCs) and precursors for several lineages[1,2]. Gfi1b is expressed in HSCs, myeloid/erythroid precursors (MEPs), megakaryocytes (MKs) and to varying levels during erythrocyte maturation[2]. Both Gfi1 and Gfi1b have an N-terminal Snail/Gfi1 (SNAG) domain, which enables transcriptional repression through the recruitment of cofactors Lysine (K)-specific demethylase 1A (LSD1/KDM1A) and CoREST/Rcor1[3–5]. Interestingly, LSD1 seems to play an essential structural rather than enzymatic role as part of the Gfi1 repressive complex[6]. LSD1's loss disrupts Gfi1's association with and repression of target loci. Gfi1 and Gfi1b also both interact with co-repressors, such as histone lysine methyltransferase 2 (EHMT2/G9a) and histone deacetylases (HDAC1/2)[4,7,8]. While germline deletion of Gfi1b in mice causes lethality at around day 14.5 of embryonic development, conditional knockout mice have been generated and show that Gfi1b controls HSC and MK expansion[9,10]. While Gfi1b-deficient HSCs remain functional and give rise to all hematopoietic lineages upon transplantation, MKs that lack Gfi1b cannot produce platelets and are unable to respond with spreading and membrane ruffling to integrin receptor stimulation due to defects in cytoskeletal organization[11].

Wnt/β-catenin signaling also plays a crucial role in early hematopoiesis, notably in HSCs. Loss- and gain-of-function studies demonstrated that tight control of Wnt signaling and β-catenin activity is necessary for proper function and cellularity control of hematopoietic cells including HSCs and MKs[12–15]. Overactive Wnt/β-catenin signaling leads to exhaustion of HSCs, but insufficient activation is equally detrimental[16,17]. β-catenin acts as a transcriptional co-activator in complexes with transcription factors, such as the T-cell factor/lymphoid enhancer factor (TCF/LEF) family members to regulate gene expression. The canonical Wnt signaling is under negative regulation at various levels. For instance, GRG/TLE (Groucho/transducin-like enhancer) proteins associate with TCF molecules in the nucleus to switch off expression of Wnt target genes in the absence of nuclear β-catenin[18]. CtBP1 and HDACs are other negative regulators of canonical Wnt signaling. Multiple non-canonical Wnt signaling pathways also exist and although these pathways all function in a β-catenin independent manner, crosstalk exists between canonical and non-canonical signaling pathways in various contexts[19,20]. Several studies have shown that non-canonical Wnt signaling antagonizes the canonical Wnt pathway through different mechanisms[21,22]; one example being NFAT5, which is a transcription factor downstream of the non-canonical Wnt/Ca$^{2+}$ pathway that inhibits canonical Wnt signaling via inhibition of β-catenin acetylation[21].

Here we present evidence that Gfi1b controls HSC and MK cellularity and MK spreading in response to integrin substrates by regulating Wnt/β-catenin signaling. Our results show that Gfi1b interacts with β-catenin as well as regulators of Wnt/β-catenin signaling pathway and that loss of Gfi1b affects the expression of Wnt target genes in both MKs and HSCs. We also reveal a tripartite Gfi1b/LSD1/β-catenin complex that co-occupies key Wnt/β-catenin signaling target regions like the *Axin2* promoter. We show that Gfi1b can enhance transcription of TCF/LEF dependent promoters and reporter genes in vitro and in vivo and we present evidence that Gfi1b does this by recruiting LSD1 via its SNAG domain to β-catenin containing complexes. In agreement with this, we show that Gfi1b-deficient HSCs and MKs have decreased levels of canonical Wnt signaling in vivo, which can be reversed when Wnt/β-catenin signaling is stimulated externally by Wnt3A treatment.

## Results

### Gfi1b deficiency leads to expansion of HSCs and MKs.
To generate Gfi1b-deficient (KO) mice we introduced a *Rosa-CreER* transgene into *Gfi1b^{fl/fl}* mice[9,11]. Floxed *Gfi1b* alleles were deleted by tamoxifen injections (Fig. 1a) and confirmed in both MKs and HSCs by the absence of floxed exons 2–4 expression (Supplementary Fig. 1a). MKs and their progenitors were defined as lin⁻, CD4$^{high}$, CD9$^{high}$ cells, and HSCs were defined either as lin⁻, Sca-1⁺, cKit⁺ (LSK), CD48⁻, CD150⁺, CD41$^{high}$/CD9$^{high}$ or CD41$^{low}$/CD9$^{low}$ to eliminate potential contamination by MK precursors (Supplementary Fig. 1b), and also because we had shown previously that the expression of integrin molecules, such as Itga2b (CD41) and Itgb3 (CD61), are up-regulated in Gfi1b-deficient HSCs[9]. Following the ablation of Gfi1b, we observed increased numbers of MKs and CD41$^{low}$/CD9$^{low}$ HSCs in the bone marrow and blood, (Fig. 1b, c) confirming previous results[11]. Moreover, Gfi1b-deficient MKs and HSCs showed the same expansion in vitro after Tamoxifen injection (Fig. 1d, e) supporting that Gfi1b controls the cellularity of both HSCs and MKs in bone marrow and in peripheral blood in a cell autonomous manner[9].

We recently showed that Gfi1b loss leads to impaired MK spreading and motility on fibronectin and other matrix substrates[11]. On such substrates, Gfi1b KO MKs are smaller, stay completely round, and do not form protrusions like WT cells or form platelets (Fig. 1f)[11]. A comparison of the roundness index (RI) of cultured cells clearly showed that Gfi1b-deficient MKs maintained significantly low RIs, indicating that they are unable to spread and form protrusions, confirming a poor integrin ligands response (Fig. 1g).

### GFI1B is associated with regulators of Wnt pathway.
To understand how Gfi1b deficiency leads to expansion of HSCs and MKs and loss of MK spreading, a Flag-tagged version of GFI1B was expressed in HEK293 cells to identify interacting proteins by immune-precipitation and mass spectrometry. In addition to proteins already known to bind to GFI1B, such as LSD1, HDACs, CoRest factors and G9a[4], we found β-catenin and several other proteins with regulatory roles in the canonical Wnt pathway as new potential GFI1B binding partners (Fig. 2a, Supplementary Fig 1c, Supplementary Table 2). These included the DNA helicase and chromatin remodeling factor CHD8, which silences β-catenin mediated transcription[23], Pontin52 (also known as TIP49), a c-Myc interacting protein involved in chromatin remodeling and transcription[24], APC, a tumor suppressor and antagonist of the Wnt signaling pathway[18], PP2A, a phosphatase that regulates Wnt signaling at several levels[25,] and CtBP1 (C-terminal binding protein1), a repressor of Wnt signaling that associates with LSD1[26] (Fig. 2a, Supplementary Fig 1c). Gene Ontology (GO) enrichment analysis confirmed that numerous GFI1B associated proteins were annotated under the biological process positive regulation of Wnt signaling pathway (Supplementary Fig 1d).

Similarly, a GFI1B-BirA* BioID approach showed that regulators of the Wnt pathway, such as KDM2A and KDM2B, which regulate the stability of nuclear β-catenin via demethylation[27], TLE1 and TLE3 which are both β-catenin inhibitors[28] and CREBBP (also known as CBP) a histone acetyltransferase that binds to β-catenin[29] (Fig. 2b, Supplementary Fig 1e) can interact with GFI1B. In addition, we also found Spindlin1, a chromatin reader that prompts Wnt signaling[30], UBR5, an E3 ubiquitin-protein ligase that inactivates TLE[31], CDC73, which is a β-catenin interacting tumor suppressor[32], CCAR2, which enhances LEF1-β-catenin complex formation[33] and BRG1, a chromatin remodeler involved in transactivation of Wnt target genes (Fig. 2b)[34].

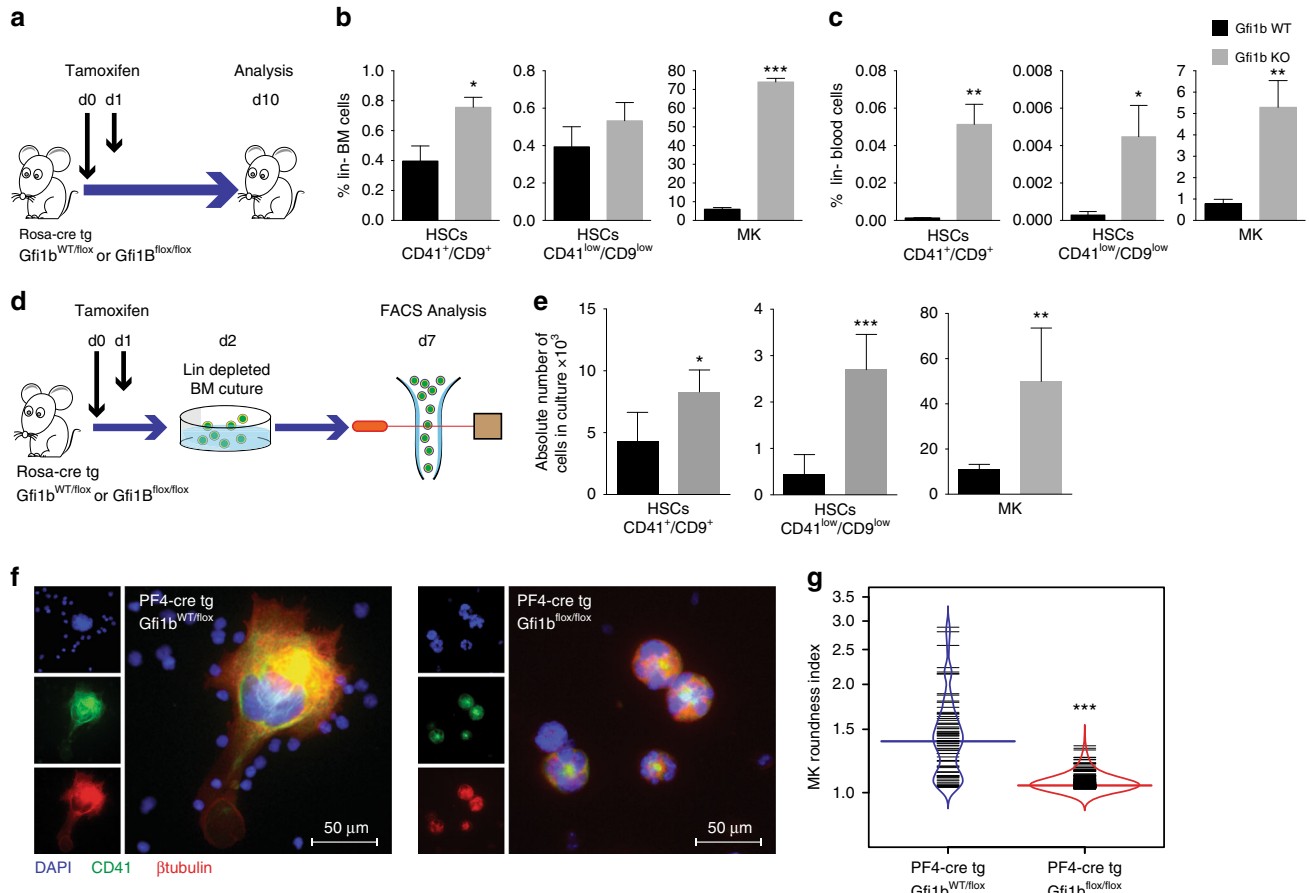

**Fig. 1** Loss of Gfi1b in MKs and HCSs causes their expansion and mobilization into the blood. **a** Schematic of Tamoxifen injections; mice were analyzed by FACS at day 10 following the first injection. **b**, **c** Quantification of Gfi1b WT/KO MKs and HSCs in lineage negative depleted BM and blood ($n = 3$ mice per group). **d** Schematic of in vitro analysis of Gfi1b WT/KO HSCs and MKs expansion. **e** MKs and HSCs (CD41[low]/CD9[low]) were taken into culture and were quantified by FACS at day six (day seven post Tamoxifen injection, $n = 3$ mice per group). **f**, **g** Loss of Gfi1b leads to impaired MKs spreading on Fibronectin coated matrix as quantified by calculating their roundness index ($n = 70$ Gfi1b wt/flox cells and 220 Gfi1b flox/flox cells). (*$p < 0.05$, **$p < 0.001$, ***$p < 0.0001$ on a Welch corrected $t$-test, error bars show s.d)

We noticed a well-conserved six amino-acid motif in the N-terminal sequence of GFI1B protein that we named WRD for Wnt regulatory domain, since a similar motif has previously been shown in other proteins to mediate binding of the β-catenin inhibitors TLE1 and TLE3[28] (Fig. 2c). Immune-precipitation of Flag-tagged WT GFI1B or different mutants, in which either the SNAG, WRD or intermediate domain were disrupted, confirmed that GFI1B interacts with β-catenin, CHD8, Pontin52, and CtBP1 and to some extent with both TLE3 and TLE1 (Fig. 2d). The SNAG domain was required for an interaction between GFI1B and CtBP1 and, as previously shown, LSD1[4], whereas the WRD domain was necessary for GFI1B interaction with CHD8, TLE3, Pontin52, and to some extent TLE1 (Fig. 2d). Moreover, GFI1B's zinc finger domain was required for its interaction with β-catenin (Fig. 2e). Endogenous immune-precipitation with extracts from non-transfected K562 and HEL cells demonstrated an enrichment of complexes between GFI1B, β-catenin, and Pontin52 (Fig. 2f–h and Supplementary Fig. 2a). These results indicate that GFI1B interacts with β-catenin and several Wnt/β-catenin signaling regulators.

β-catenin co-precipitated with GFI1B and LSD1 and increased levels of LSD1 protein were collected when exogenous WT GFI1B was present (Fig. 3a). This was not true when the GFI1B P2A mutant was present instead, which is unable to bind LSD1[5] (Fig. 3a). Comparable results were obtained with anti-β-catenin

immune-precipitations in the presence of GFI1B or the GFI1B ΔSNAG mutant that lacks the 20 N-terminal amino acids that are required for LSD1 binding (Fig. 3b). Similarly, immune-complexes collected with GST-E-Cadherin that binds to β-catenin from U2OS cells engineered to express doxycycline-inducible GFI1B only contained more LSD1 when GFI1B expression was induced (Fig. 3c), suggesting that GFI1B recruits LSD1 to β-catenin or β-catenin-containing protein complexes. To characterize these complexes further, we performed β-catenin-BirA* BioID in the absence or presence of WT GFI1B or two GFI1B mutants (Fig. 3d, e). All three GFI1B forms were biotinylated by β-catenin-BirA* further indicating an interaction or close proximity between GFI1B and β-catenin (Fig. 3d, e) and confirming both our mass spectrometric and co-IP experiment findings that also suggest that GFI1B binds to β-catenin (Fig. 2a, d, and e).

**GFI1B enhances transcription of TCF/LEF dependent reporter.** We found that GFI1B enhanced transcription of a *TCF* promoter reporter both at basal level and in the presence of Wnt3A, a canonical Wnt signaling ligand, following over-expression of Wnt3A or the active form of β-catenin or upon treatment with LiCl, a GSK3β inhibitor that stabilizes β-catenin (Fig. 4a–e), or CHIR99021, a specific GSK3β inhibitor (Fig. 4f, Supplementary Fig. 2b, c) indicating that GFI1B can activate β-catenin/TCF

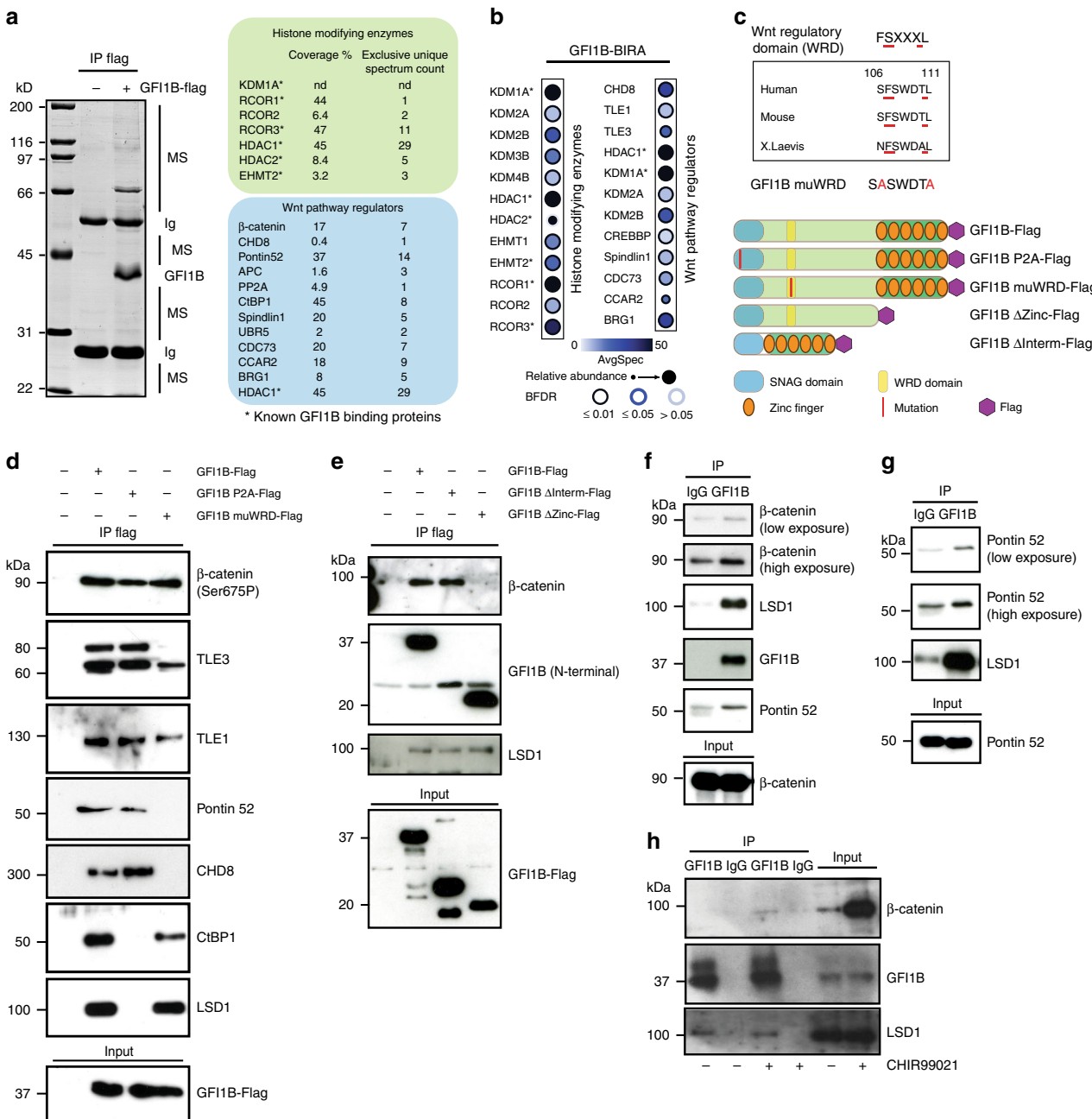

**Fig. 2** GFI1B interacts with β-catenin and several regulators of the Wnt/β-catenin signaling pathway. **a** Immune precipitation (IP) with anti Flag antibody from 293T cells overexpressing GFI1B-Flag followed by mass spectrometric analysis. **b** Dot Plot showing BioID interactions of GFI1B-BirA*-Flag with the indicated Histone modifying enzymes and Wnt pathway regulators in Flp-In T-REx HEK293 cells. Node color represents the average spectral counts. The edge color depicts the confidence score of the interaction (BFDR ≤ 1% as high confidence, 1% < BFDR ≤ 5% as medium confidence or 5% < BFDR as low confidence score). The relative abundance of prey across the bait is represented by the size of the circle. **c** A previously described Wnt regulatory domain (WRD) is present in GFI1B[28]. The indicated Flag-tagged and mutated forms used for IP. **d, e** Immune precipitation with anti Flag antibodies from 293T cells overexpressing WT or the indicated mutated forms of GFI1B followed by western blot. **f–h** Immune precipitation with an anti GFI1B antibody from K562 (**f**), HEL (**g**), and CHIR99021 treated K562 (**h**) cells

dependent transcription. This GFI1B activity was reduced when its SNAG (GFI1BP2A or GFI1BΔSNAG), WRD (GFI1BΔWRD), intermediate or zinc-finger domains were disrupted, either in the presence of LiCl or CHIR99021 (Fig. 4c, f). We observed the same reduction in GFI1B activity using single (GFI1BF106A or GFI1BL111A) or double mutations in the WRD sequence (GFI1B-muWRD, depicted in Fig. 2c) (Fig. 4d, e). This indicates that the observed effect of GFI1B on β-catenin/TCF mediated transcription depends on its ability to bind LSD1 or CtBP1 and/or other

β-catenin regulators that interact with β-catenin through these domains. In agreement with this, GFI1B was more active in combination with LSD1 than alone (Fig. 4g), suggesting that the GFI1B/LSD1 complex can be an activator of β-catenin/TCF mediated transcription in this system. The repressive GFI1B activity on its own promoter was not affected by WRD sequence deletion or CHIR99021 treatment (Supplementary Fig. 2b). Furthermore, we observed that GFI1B was able to partially reverse the CtBP1 and TLE1 inhibitory effect on β-catenin/TCF-dependent

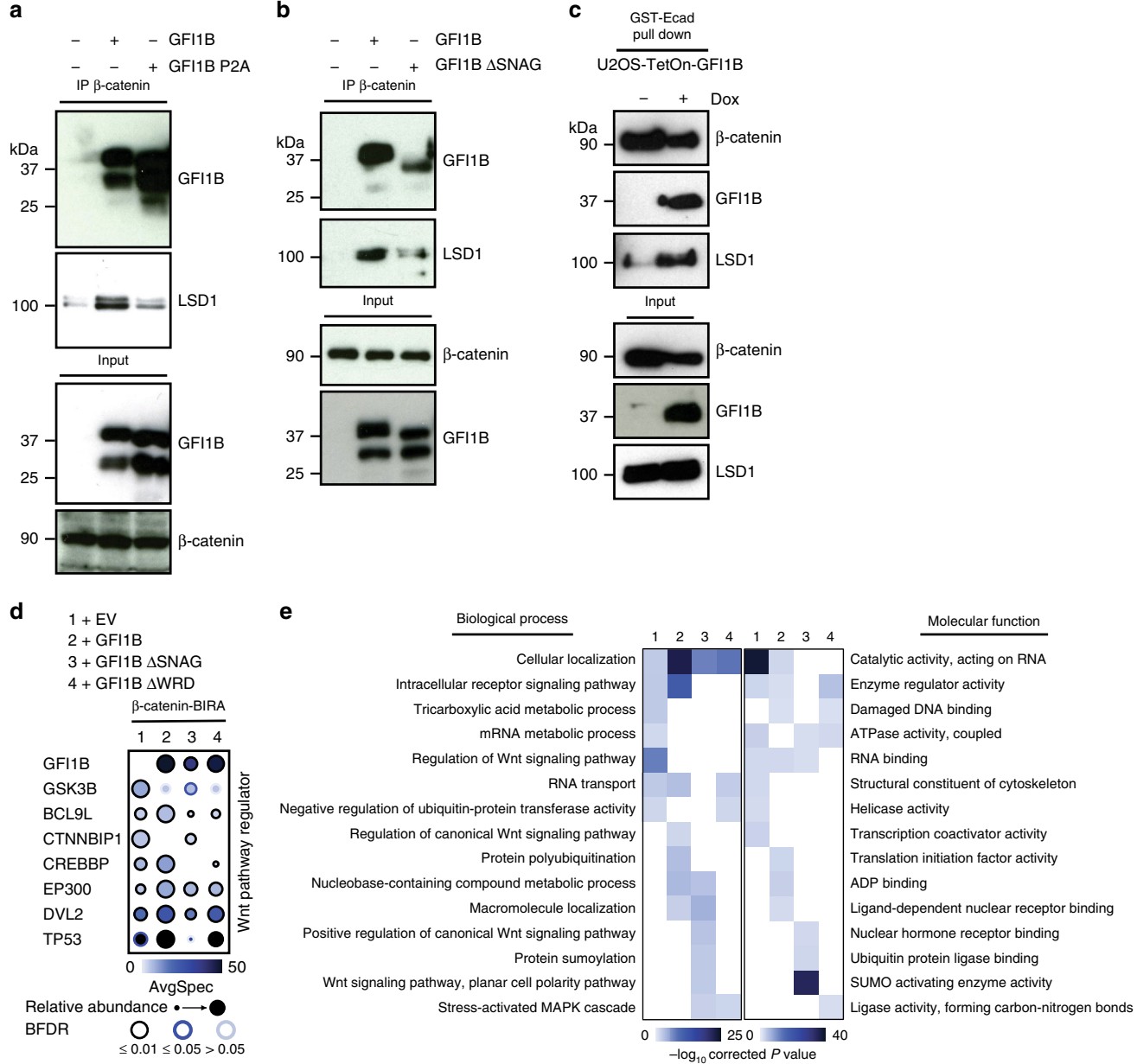

**Fig. 3** GFI1B recruits LSD1 to β-Catenin containing complexes. **a**, **b** Western blot analysis of indicated proteins after immune precipitation of endogenous β-catenin from 293T cells transfected with the indicated GFI1B constructs. **c** GST-E-Cadherin pull down of endogenous β-catenin in U2OS cells expressing inducible GFI1B (Tet-ON system) followed by Western blot analysis. **d** Dot Plot showing BioID interactions of β-catenin-BirA*-Flag coexpressing an Empty Vector (1), the wild-type form of GFI1B (2), GFI1B lacking the SNAG domain (3) or GFI1B lacking the WRD domain (4) with the indicated Wnt pathway regulators in Flp-In T-REx HEK293 cells. **e** Heat map illustrating biological processes and molecular functions of associated GO terms. The enrichment score of each GO term is shown as the −log10 of corrected $P$ values, indicated by different color intensities

transcription (Fig. 4h). Similarly, inducible GFI1B expression under doxycycline in U2OS cells reversed the effect of TLE1 in the TOP/FOP reporter assay (Fig. 4i). This suggests that GFI1B can also activate β-catenin/TCF transcription by counteracting the effect of β-catenin inhibitors.

Neither TOP nor FOP constructs contain a known GFI1B consensus-binding motif (AATC). This suggested that GFI1B activates TCF/LEF-dependent transcription without binding DNA directly. To test this, we generated TOP reporter genes with GFI1B binding motifs (AAATCTCTGCA) before and after the TCF/LEF binding sites and observed that GFI1B was now able to inhibit transcription of this modified TOP reporter (Fig. 4j). This inhibition was dependent on GFI1B DNA-binding capacity since a N290S mutation of GFI1B's fifth zinc-finger domain,

which abolishes DNA-binding of GFI1B, did not show any inhibition in this modified TOP reporter assay but rather activation similar to WT GFI1B (Fig. 4j). GFI1B N290S mutant in the unmodified TOP assay could still associate with β-catenin and activate TCF/LEF mediated transcription to the same extent as WT GFI1B (Fig. 4k, l), indicating that the ability of GFI1B to activate TCF-dependent transcription can be independent of its ability to bind DNA and by inference also its function as a transcriptional repressor.

**Gfi1b KO deregulates expression of Wnt target genes**. To test whether this GFI1B activity can also be observed in vivo, we used a well-established $Axin2^{LacZ}$ knockin reporter mouse, which enables monitoring of TCF/LEF dependent transcription by

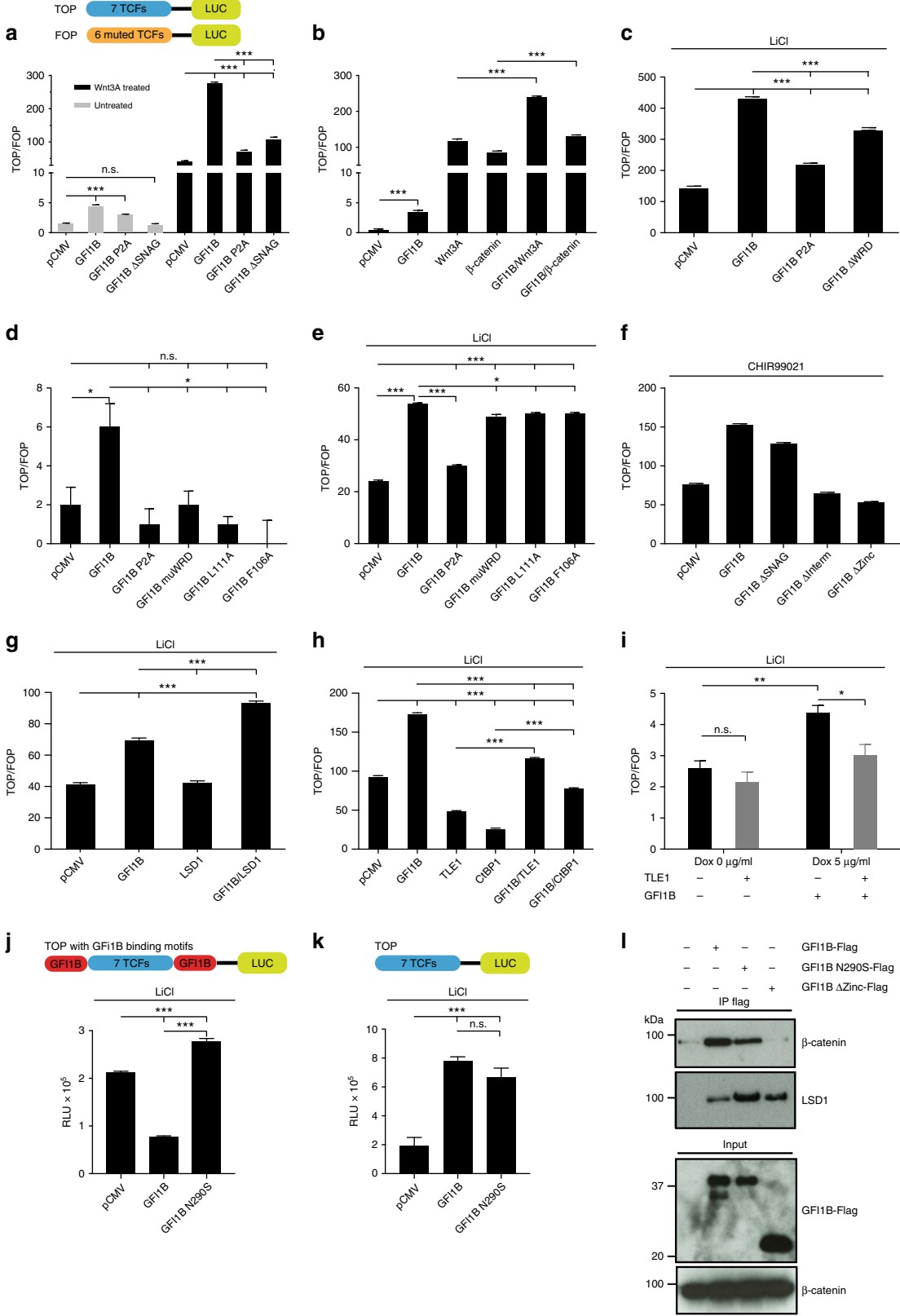

measuring β-galactosidase (LacZ) activity in mouse primary cells (Fig. 5a, Supplementary Fig. 2d)[35]. Compared to controls, β-galactosidase (LacZ) activity was reduced in HSCs and MKs with Gfi1b deletion (Fig. 5b). Quantification of mean fluorescence intensities (MFI) of LacZ in these cells showed the effect of Gfi1b deficiency on the *Axin2* promoter activity was measurable in all

three cell subsets and was statistically significant in MKs (Fig. 5c). To validate this, we used a second in vivo reporter system, the *TCF/LEF:H2B-GFP* transgene reporter mice, in which six copies of a *TCF/LEF* responsive element are placed directly 5′ of a histone *H2B-GFP* fusion gene[36]. Similar to the *Axin2*[LacZ] reporter system, a reduction of GFP intensities was observed in HSCs and

**Fig. 4** GFI1B enhances transcription of a TCF dependent promoter/reporter system. **a**, **b** TOP/FOP flash reporter assay in 293T cells. Cells were transfected with the indicated plasmids together with TOP or FOP reporter constructs 36 h before luciferase measurement. GFI1B enhances the TOP/FOP ratio at both basal levels and following activation with **a** 10% Wnt3A-conditioned media or **b** co-transfection with active form of β-catenin and/or Wnt3A expressing vectors. **c**–**e** TOP/FOP flash reporter assay in 293T cells transfected with WT Gfi1b and the indicated mutated forms. Cells were treated with 25 mM LiCl (to activate canonical Wnt signaling in **c** and **e**) for 5 h before luciferase measurement. **f** TOP/FOP flash reporter assay in 293T cells treated with CHIR99021, a specific GSK3β inhibitor for 5 h before luciferase measurement. **g**, **h** TOP/FOP flash reporter assay in 293T cells and GFI1B-TetON-U2OS cells transfected with the indicated vectors. **i** TOP/FOP flash reporter assay in U2OS cells stably expressing a doxycycline inducible GFI1B (Tet-ON system) transfected or not with a construct for TLE in the presence or absence of doxycycline. **j**, **k** TOP and modified TOP (flanked by GFI1B binding motifs) reporter assay in 293T cells. **l** Immune precipitation with anti Flag antibodies from 293T cells overexpressing WT or the indicated mutated forms of GFI1B followed by western blot. (*$p < 0.05$, **$p < 0.005$, ***$p < 0.0001$ on a Welch corrected $t$-test, error bars show s.d, $n = 3$ biologically independent samples for each data point)

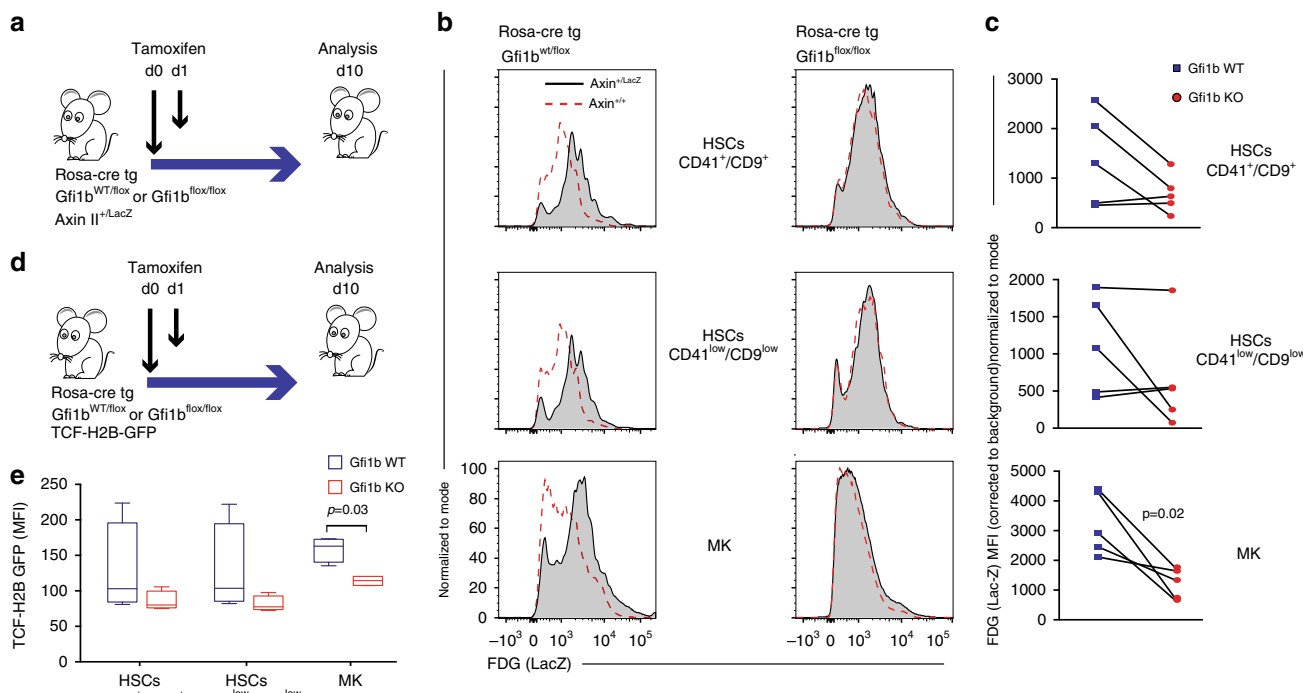

**Fig. 5** Gfi1b is required for the expression of Axin II- or TCF-dependent reporter alleles in HSCs and MKs in vivo (**a**–**c**) Axin2$^{+/LacZ}$ Wnt-reporter mice were used. Littermates not carrying the reporter transgene (Axin2$^{+/+}$) were used to correct for endogenous β-galactosidase activity. **a** Mice were analysed by FACS at day 10 following the first injection of tamoxifen. **b** FDG (LacZ staining) histograms of indicated FACS gated cells. A representative result from three independent experiments is shown. **c** Quantification of the mean fluorescence intensity (MFI) measured in the FDG channel (LacZ) normalized to endogenous β-galactosidase activity. Paired t-test was used for the analysis. ($n = 5$ mice per group) (**d**, **e**) In vivo measurement of canonical Wnt signaling activity in Gfi1b WT/KO MKs and HSCs by FACS using TCF-H2B-GFP transgene reporter mice. A Mann–Whitney U non-parametric test was used for the analysis. Boxplot Centre line shows median and bounds of box and whiskers show interquartile ranges of GFP MFI, $n = 5$ GFI1b WT and five GFI1B KO mice

MKs following Gfi1b deletion, again most significantly in MKs (Fig. 5d, e). These data indicate that Gfi1b absence leads to a transcriptional down-regulation of β-catenin driven target genes in vivo.

RNA-seq analysis comparing flow-sorted Gfi1b WT/KO CD41$^+$/CD9$^+$ HSCs, MKs, and CD41$^{low}$/CD9$^{low}$HSCs showed that many genes, which were differentially expressed between WT and Gfi1b-deficient cells, belong to previously published sets of genes[25] involved in Wnt/β-catenin signaling (Fig. 6a, b). A set of genes that were described to be up-regulated following activation of canonical Wnt signaling[37] was enriched in Gfi1b WT CD41$^+$/CD9$^+$ HSCs and CD41$^{low}$/CD9$^{low}$HSCs and a set of genes down-regulated upon activation of canonical Wnt signaling, was enriched in Gfi1b KO MKs (Fig. 6c).

Next, we identified genes that were expressed over 1.5 fold-up or 0.6 fold-down in MKs, HSCs and CD41$^-$/CD9$^-$ HSCs and were also co-occupied by Gfi1b and β-catenin or also by LSD1

(Fig. 6d, Supplementary Table 2, Supplementary Fig. 3). To obtain this information, we interrogated published ChIP-seq datasets from the mouse embryonic stem cell (ESC)-derived hematopoietic precursor cell line HPC-7 for Gfi1b[38], mouse ESCs expressing a tagged-β-catenin[39] and with wt ESCs for LSD1[40]. We found 523 genes co-occupied by Gfi1b and β-catenin and 133 genes co-occupied by all three factors at promoters (Supplementary Fig. 4a, Supplementary Table 3). Among them were known canonical Wnt genes that were down-regulated in Gfi1b-null cells such as Ccnd1 (regulator of cell cycle progression)[41], Egr1 (transcription factor)[42] and Ptgs2 (enzyme in prostaglandin biosynthesis)[43] (Supplementary Figs 3 and 4b, Fig. 6d). However, key non-canonical Wnt genes such as Ptk7 (Wnt co-receptor)[22], Nfat5[21], Zfp467 (regulator of the canonical Wnt inhibitor SOST)[44], and Rock2 (serine/threonine kinase downstream of PTK7)[45] were up-regulated in Gfi1b-deficient MKs (Supplementary Table 2, Supplementary Figs 3 and 4b).

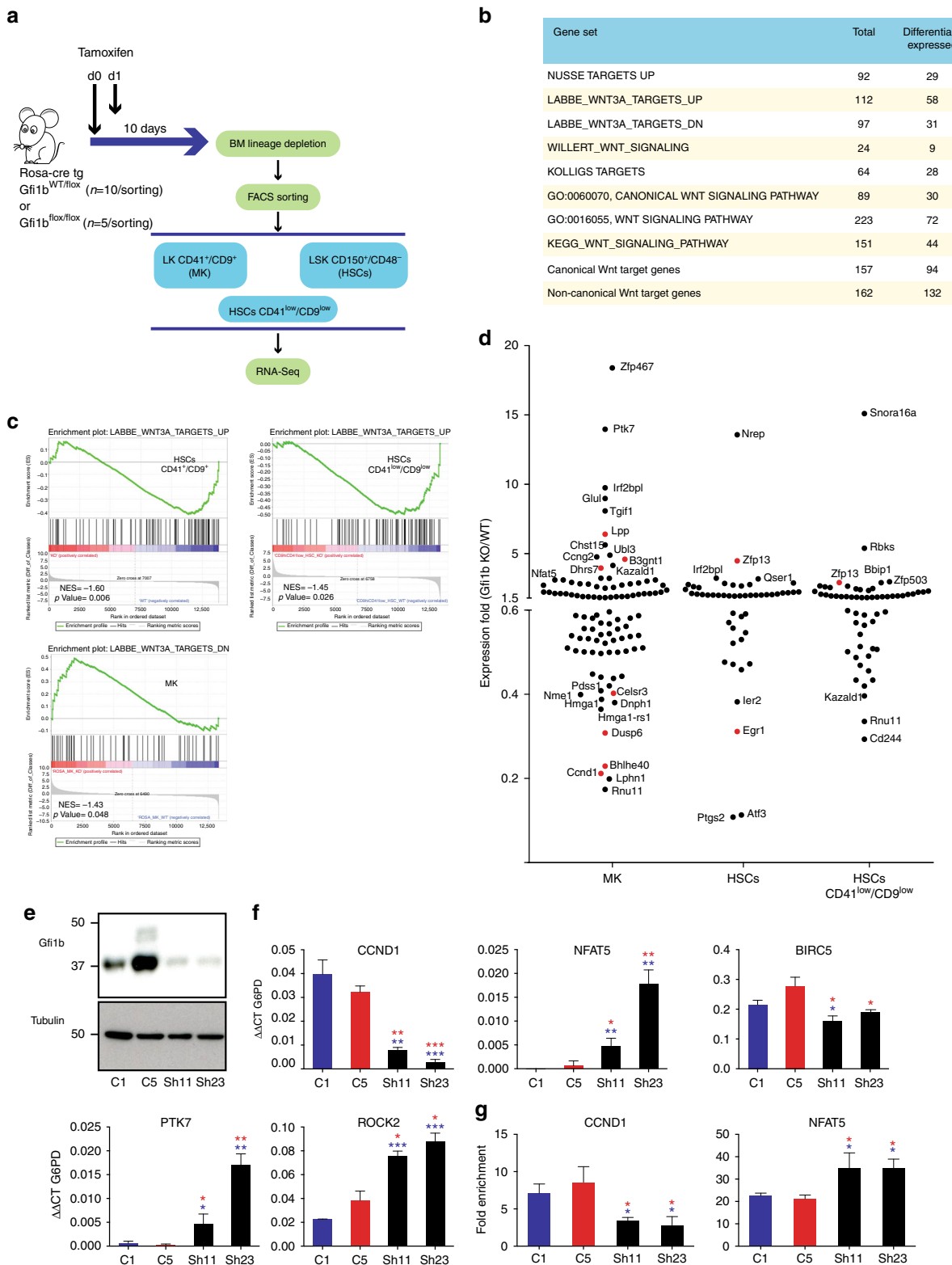

**Fig. 6** Deregulation of Wnt/β-catenin target gene expression in Gfi1b-deficient MKs and HSCs. **a** Gene expression analysis was done on sorted Gfi1b WT/KO MKs, HSCs and HSCs CD41^low/CD9^low as indicated. **b**, **c** Gene set enrichment analysis (GSEA) of RNA-seq data from the sorted cells using published Wnt related gene sets (see Supplementary Table 2 for gene set references). **d** RNA-seq expression analysis of Gfi1b/β-catenin co-occupied genes (based on published ChIP-seq data) in the three Gfi1b WT/KO populations. Genes indicated in red also have peaks for LSD1 at their promoter. **e** GFI1B expression in two independent K562-derived cell lines stably expressing a scrambled control shRNA (C1, C5) and two lines expressing a GFI1B specific shRNA (Sh11, Sh23). **f** Expression analysis of Wnt target genes by RT-PCR in GFI1B shRNA KD and scramble shRNA stable K562 clones. **g** Enrichment of acetylated H3K9 at GFI1B/β-catenin target gene promoters by ChIP-qPCR in GFI1B shRNA KD and scramble shRNA stable K562 clones. (*$P < 0.05$, **$P < 0.001$, ***$P < 0.0001$ on a Welch corrected $t$-test compared to C1 (in blue) or C5 (in red), error bars show s.d, $n = 3$ technical replicates)

To validate our findings further, we generated a GFI1B knockdown using shRNA in the human erythroleukemia cell line K562 (Fig. 6e). We confirmed reduced GFI1B levels decreased the expression of *CCND1* and *BIRC5* (canonical Wnt genes) and increased the expression of *NFAT5, ROCK2* and *PTK7* genes (non-canonical Wnt genes) (Fig. 6f), similar to that observed in cells from Gfi1b-deficient mice (RNA-seq data Fig. 6d, Supplementary Fig 3, Supplementary Table 2). Furthermore, the regulation of *CCND1* or *NFAT5* expression in GFI1B knockdown K562 cells correlated with lower or higher H3K9 acetylation at their promoters, respectively (Fig. 6g). This suggests that the expression changes induced by GFI1B correlate with the appropriate H3K9 acetylation changes at promoters of target genes.

Next, we performed RNA-seq analysis on two sets of Gfi1b WT and KO MKs obtained by cultivating lin-BM cells in vitro with TPO and SCF for 7 days (Fig. 7a). As with the previous RNA-Seq, many genes that were differentially expressed between WT and KO samples were part of previously published genes sets involved in Wnt/β-catenin signaling (Fig. 7b, c)[25]. Especially, two canonical Wnt gene sets were significantly down-regulated in Gfi1b KO MK samples (Nusse Targets up and Labbe Targets up, Fig. 7d). Interestingly, unsupervised clustering of canonical or non-canonical Wnt target genes clearly separated Gfi1b KO and WT MKs in both biological replicates (Fig. 7e). These findings confirmed that Gfi1b loss in HSCs or MKs can lead to both up- and down-regulation of Wnt target genes.

**β-catenin co-occupies regions targeted by Gfi1b and LSD1.** The analysis of published ChIP-seq datasets from HPC7 cells for Gfi1b and knock-in ES cells for β-catenin[38–40,46] showed that Gfi1b and β-catenin co-occupy several loci within 1 kb of a promoter (Supplementary Fig. 5a and b). We next examined the genomic features found closest to a binding site for one factor as a function of its proximity to a binding site of the other factor using these datasets and a CTCF ChIP-seq dataset[46] as a control, which showed no relation with features associated with Gfi1b and only a limited relation for β-catenin. When Gfi1b peaks overlap a β-catenin peak, they are more frequently present at transcription start sites (TSS) and conversely, β-catenin peaks are more likely found at a TSS when they overlap a Gfi1b peak, as compared to peaks with more than 1 kb distance to the other factor (Supplementary Fig. 5a–c). A comparable pattern was found for both Gfi1b and β-catenin peaks relative to their distance to LSD1 binding sites (Supplementary Fig. 5b and c).

To confirm a co-occupation of GFI1B, β-catenin, and LSD1 at specific genomic sites at endogenous expression levels in the same cells, we performed ChIP-seq analysis with K562 cells treated or not with Wnt3A for 4 h to obtain the highest levels of active, nuclear β-catenin (Supplementary Fig. 6). β-catenin peaks showed a statistically significant overlap with GFI1B and LSD1 peaks at promoters (Fig. 8a, Supplementary Fig. 7a and Supplementary Table 4) and also that many more genes are occupied by the tripartite complex (GFI1B/β-catenin/LSD1) than by LSD1/β-catenin or by GFI1B/β-catenin (Fig. 8a). *AXIN2* and *YAF2* are examples of gene promoters specifically targeted by β-catenin, GFI1B, and LSD1 upon Wnt3a treatment (Fig. 8b).

Motif analysis revealed, as expected, that sites bound by GFI1B were significantly enriched for the GFI1B binding motif as well as GATA binding motifs genome-wide (Fig. 8c). However, sequences at sites co-occupied by β-catenin and GFI1B were found to be prominently enriched for the LEF1 motif as well as GATA factor motifs, but not for GFI1B's own binding motif (Fig. 8c). This suggests that GFI1B has a different DNA binding activity specifically at β-catenin sites. Notably, Wnt stimulation

did not alter the majority of GFI1B and LSD1 targeted promoters and enhancers (Supplementary Fig. 7a, b). β-catenin bound most frequently to promoters and enhancers where GFI1B and LSD1 are already present prior treatment (Supplementary Fig. 7a, b).

To assess whether recruitment of β-catenin is GFI1B dependent, we carried out both GFI1B and β-catenin ChIP-qPCRs on CHIR99021 treated K562 cells with WT (K562-C5) and KD (K562-S23) levels of GFI1B. Levels of GFI1B enrichment confirmed the decrease in GFI1B protein at target genes within GFI1B KD K562 cells (Fig. 8d). Importantly, depletion of GFI1B lead to a significant decrease in β-catenin enrichment at target loci, such as *SLC38A8* and *YAF2*, indicating that β-catenin recruitment is GFI1B-dependent at these promoters (Fig. 8d). Other loci, like *AXIN2* and *SP5*, did not show significant changes in β-catenin levels with GFI1B depletion, indicating that β-catenin may also bind to specific loci independently of GFI1B in K562 cells.

Since LSD1 is a histone demethylase, specifically for histone 3, lysine 4, and 9 (H3K4 and H3K9), we assessed H3K4me1 and H3K9me2 levels by ChIP-seq in K562 cells before and after Wnt3a treatment. The majority of genes have high H3K4me1 enrichment if targeted by one or more of β-catenin, GFI1B and LSD1 upon treatment (60–90%). Only when none of the three factors were detected at promoters did similar proportions of genes experience increases or decreases in H3K4me1 levels (Supplementary Fig. 7c). Furthermore, promoters targeted by one or more of the three factors mostly had a distinct decrease in H3K9me2 enrichment (60–100%). This indicates that upon Wnt stimulation chromatin priming may occur with a general increase in H3K4me1 and decrease in H3K9me2 at promoter targets of one or more of the three factors and not due to LSD1 alone.

**Genomic distribution of β-catenin, GFI1B and LSD1 peaks.** GFI1B associates mostly at promoter regions (~48%), while LSD1 was located primarily within intergenic (~37%) or intronic (~41%) regions (Supplementary Fig. 7d). This genomic distribution did not significantly change upon Wnt3A stimulation. β-catenin was bound within intergenic regions mostly (~61.5%), however, upon treatment, the most notable shift appears as a 55.7% decrease in the proportion of β-catenin peaks at promoters. The genomic distribution for one factor as a function of its proximity to a binding site of another factor showed that β-catenin peaks overlapping with a GFI1B or LSD1 peak shift from predominantly promoter (61 to 24.5%) to intergenic regions (26.5–50%) upon Wnt3A activation (Fig. 8e). Analysis of established enhancer regions showed there was a strong increase in the number of enhancers co-bound by β-catenin, GFI1B, and LSD1 following Wnt activation (Fig. 8f–h and Supplementary Table 5). Similarly to promoters, a shift was seen mostly with β-catenin localization rather than GFI1B or LSD1 at enhancers with Wnt3a treatment (Supplementary Fig. 7b).

**Increased Wnt/βcatenin signaling rescues Gfi1b KO phenotype.** To test whether Gfi1b null phenotypes can be rescued by reactivating Wnt/β-catenin signaling pathway in primary HSCs and MKs, we explanted bone marrow cells from Rosa Cre-ER, Gfi1b[fl/fl] mice or controls after injection of Tamoxifen to induce Gfi1b deletion, followed by a lineage depletion. The cells were put in culture and treated with recombinant Wnt3A (a canonical Wnt ligand) (Fig. 9a). After 7 days in culture, non-treated HSC and MK samples from Gfi1b KO mice expanded as expected several fold over WT controls (Fig. 9b). However, treatment with Wnt3A inhibited this expansion in a dose dependent manner, most significantly in MKs (Fig. 9b). To validate this hypothesis further, we infected isolated CD45.2+ Gfi1b-deficient lin-BM cells with

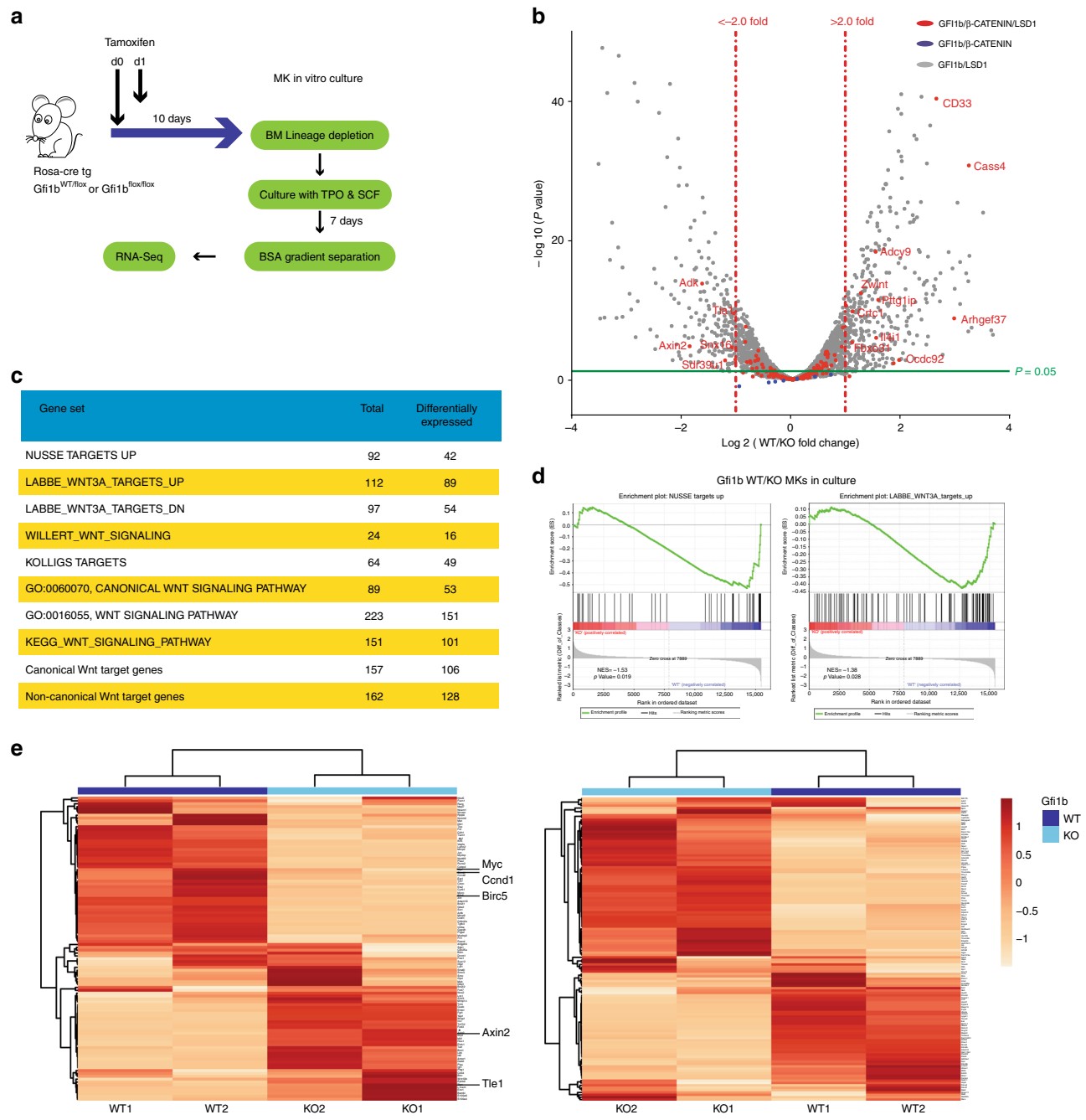

**Fig. 7** Canonical and non-canonical Wnt target genes are deregulated in Gfi1b KO MK in vitro. **a** Gfi1b WT/KO mice were taken to obtain lineage depleted BM cells, $n = 2$ GFI1B WT and GFI1B KO mice. Cells were taken into culture in the presence of TPO and SCF for 7 days and MK were separated using BSA gradient followed by RNA-seq analysis. **b** Volcano plot of expression fold change (KO/WT) over the $P$ value of differences between WT and KO. Vertical dashed lines show the threshold of two-fold change in expression in KO samples compared to WT. The color code shows the co-occupation of promotor by indicated factors as seen in ChIP-seq data from K562 cells. **c**, **d** Gene set enrichment analysis (GSEA) of RNA-seq data from the sorted cells using published Wnt related gene sets and two new sets of Wnt target genes pooled from published data. **e** Unsupervised clustering of RNA-seq data using indicated gene sets

retroviral vectors directing the expression of the constitutively active form of β-catenin or GFP as a control. These cells were transplanted into irradiated CD45.1 recipient mice in order to study the impact of canonical Wnt signaling activation in Gfi1b KO cells. Flow cytometric analysis, 4 months post transplantation, showed that Gfi1b-deficient β-catenin$^+$ HSCs frequencies were significantly lower than those of GFP controls (Supplementary Fig. 8a and b) suggesting that activating the canonical

Wnt signaling pathway by retroviral expression of active β-catenin, inhibits in vivo expansion of Gfi1b KO HSCs. Finally, we follow up on the findings that Gfi1b loss leads to impaired MKs spreading on fibronectin coated matrix and treat Gfi1b null MKs with Wnt3A in a dose dependent manner. The effects are quantified by measuring the RI and find that Gfi1b null MKs became less round, start spreading and resemble WT MKs when treated with Wnt3A (Fig. 9c, d). Conversely, treatment of Gfi1b

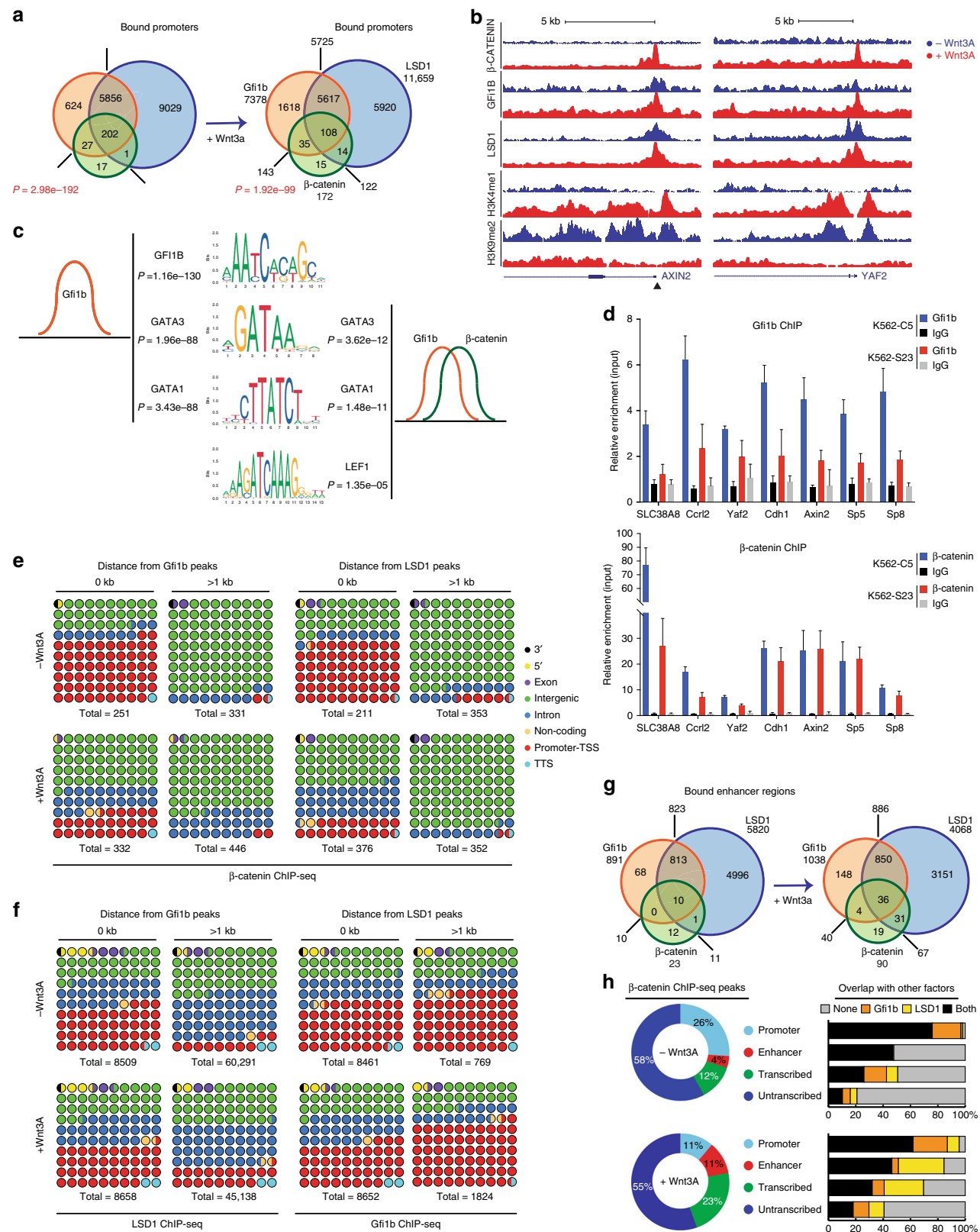

WT MKs with Wnt3A led to a RI decrease and higher Wnt3A concentration treated WT MKs resemble untreated Gfi1b KO MKs (Fig. 9d and Supplementary Fig. 8c). This data suggest that Gfi1b is required to maintain a level of Wnt/β-catenin activity necessary to control HSC and MK cellularity and the response of MKs to integrin.

## Discussion

The concept that Gfi1b regulates Wnt/β-catenin target genes is supported by several, independent lines of experimental evidence from this study: Gfi1b is found in β-catenin protein complexes found in immune-precipitations analyzed by mass spectrometry as well as BioID experiments and western blotting; Gfi1b, LSD1,

**Fig. 8** GFi1b, β-catenin and LSD1 co-occupy sites at enhancer regions and gene promoters. **a** Venn diagram showing GFI1B, β-catenin and LSD1 ChIP-seq promoter binding prior to and after Wnt3A treatment of K562 cells. *P* indicates statistical significance of overlap between GFI1B and β-catenin calculated using Fisher's exact test. **b** Example of target genes *Axin2* and *Yaf2* bound by GFI1B, β-catenin and LSD1 and H3K4me1 and H3K9me2 enrichment before and after Wnt3A treatment. Arrowhead indicates the presence of Gfi1b and TCF binding motifs in the Axin 2 promoter. **c** Motif enrichment analysis under GFI1B peaks alone (top) and Gfi1b/β-catenin overlapped peaks (bottom). **d** GFI1B (top) and β-catenin (bottom) ChIP-PCR analysis in K562 stable clones expressing GFI1B (S23) or scramble (C5) shRNA (error bars show s.d, *n* = 3 technical replicates). **e** Distribution of β-catenin peaks based on their distance to GFI1B peaks (left panel) and LSD1 peaks (right panel). **f** Distribution of LSD1 peaks based on their distance to GFI1B peaks (left panel) and reversely (right panel). **g** Venn diagram showing GFI1B, β-catenin and LSD1 binding at enhancer regions prior to and after Wnt3A treatment of K562 cells. **h** Distribution of β-catenin peaks (left) and their overlaps with GFI1B and/or LSD1 peaks based on the genomic feature of each bound region

and β-catenin co-target promoters and enhancers at endogenous levels in the erythroleukemic K562 cell line; β-catenin binds significantly less at specific target regions in K562 cells upon GFI1B depletion; GFI1B activates TCF-dependent transcription in a TOP/FOP reporter assay and Gfi1b deficiency leads to lower Wnt/β-catenin signaling in MK and HSCs in vivo when measured by two different Wnt/β-catenin dependent reporter mice. Also, Gfi1b- deficient murine HSCs and MKs show deregulated expression of Wnt target genes and knockdown of GFI1B in K562 cells reproduces this deregulation of Wnt target genes seen in cells from Gfi1b knockout mice. And lastly, cellular defects associated with Gfi1b deficiency can be in part restored by overexpressing β-catenin or by stimulating the Wnt/β-catenin signaling pathway.

Our biochemical analyses suggest that a tripartite Gfi1b/LSD1/β-catenin complex exists occupying promoter and enhancer regions. This is supported by published ChIP-seqs from transfected cells over-expressing these factors and our own ChIP-seqs on K562 cells in which this complex exists endogenously at target sites. One important line of evidence suggesting the importance of GFI1B for β-catenin recruitment to DNA is the fact that almost all β-catenin binding sites significantly overlap with GFI1B binding sites (see Fig. 8a). Notably, we discovered that GFI1B is essential for β-catenin to associate with specific target regions, like YAF2 and CCRL2, upon Wnt signaling activation. Other loci, like AXIN2 and SP5, did not show a significant change in β-catenin levels upon GFI1B depletion, despite the tripartite complex being detected at these loci in K562 cells. This indicates that β-catenin may also bind to some loci independently of GFI1B.

Interestingly, the majority of β-catenin targets are co-occupied by GFI1B and LSD1, while β-catenin only co-occupies a very small fraction of GFI1b/LSD1 targets (108 of 5727 in Wnt treated cells, see Fig. 8a). This may indicate that besides GFI1B and LSD1's influential role in Wnt/β-catenin biology, both are involved in multiple other processes, whereas many β-catenin targets are affected by GFI1B and LSD1. In addition, it is notable that while the overall number of β-catenin peaks increases following Wnt3A treatment (from 603 to 815), a considerable number of promoters were bound by β-catenin in untreated cells (247), suggesting the existence of a baseline level of activity of β-catenin.

The tripartite Gfi1b/LSD1/β-catenin complex also acts in such a way that GFI1B recruits LSD1 to β-catenin or at least enhances the interaction between LSD1 and β-catenin. This is supported by reporter gene assays and immune-precipitation experiments with mutant forms of GFI1B that lack or disrupt the SNAG domain and no longer bind to LSD1. Our data corroborate findings in other systems in particular the observation that β-catenin can recruit LSD1 to regulate the expression of the tumor suppressor Lefty1 in mouse embryonic stem cells[47]. Similarly, another study identified LSD1 as a component of β-catenin complexes that also contained DNMT1 in human colorectal cancer cell line HCT116[48]. It has been shown as well that LSD1 promotes Wnt/β-catenin pathway activation in liver cancer initiating cells and HCT 116 cell line[49,50].

We have obtained data supporting that a tripartite Gfi1b/LSD1/β-catenin complex can act as either an activator or

repressor. For instance, the canonical Wnt/β-catenin target gene *Ccnd1* promoter is occupied by Gfi1b, β-catenin, and LSD1 in mouse ESCs and its expression and H3K9 acetylation levels are decreased in Gfi1b-deficient cells upon Gfi1b knockdown. Additionally, our findings that LSD1 and GFI1B together enhance TCF mediated transcription even more than GFI1B alone support a model of a Gfi1b/LSD/β-catenin activator complex. Alternatively, we show that GFI1B interacts with a number of β-catenin inhibitors, such as CtBP1 or TLE1. It is therefore conceivable that GFI1B can also sequester these inhibitors, thereby liberating β-catenin from their inhibitory effect, leading to increased β-catenin target gene expression. The results from our TOP/FOP reporter assay with GFI1B in the presence of CtBP1 or TLE1 also support this possibility.

In contrast, the non-canonical Wnt genes *NFAT5*, *PTK7* and *ROCK2* show increased expression in Gfi1b-deficient cells and the *NFAT5* promoter shows increased H3K9 acetylation. This suggests that non-canonical Wnt targets respond differently to GFI1B than canonical Wnt/β-catenin targets. Also, regulation of Wnt/β-catenin target genes by Gfi1b is evident in primary flow sorted hematopoietic cells and MKs obtained after prolonged culturing with growth factors and cytokines. However, their Gfi1b dependent regulation of expression is not the same, suggesting that the regulatory link between Gfi1b and β-catenin and their gene expression effects is context dependent.

Analysis of consensus motifs at GFI1B and β-catenin binding sites also supports a regulatory interaction between the two proteins but do not reveal the mechanism of this interaction. While there are promoters where GFI1B and β-catenin bind only after Wnt treatment (AXIN2 and YAF2 are examples), Wnt stimulation does not alter the majority of GFI1B and LSD1 binding and β-catenin generally binds to promoters where GFI1B and LSD1 are already present. The fact that GFI1B binding motifs are less frequent at such sites compared to sites bound by GFI1B but not β-catenin suggests that GFI1B plays a role at these sites which is specific to the regulation of β-catenin targets. It should be noted that consensus motifs are not strictly required for binding to DNA. It is possible for transcription factors to bind in the absence of their established motif, and to have their nucleotide sequence preference altered by DNA shape, genomic context, DNA modifications and coding and noncoding (genetic) variation[51]. Our data suggest that GFI1B recruitment to TCF/LEF sites can be independent of its consensus element. An alternative possibility is that GFI1B can be recruited in a complex with β-catenin to TCF at sites where TCF binds DNA. In this situation GFI1B does not necessarily have to contact DNA directly and thus would not require a DNA binding motif to be present at this site. Independently of this, we also suggest an additional potential mechanism whereby GFI1B could activate TCF-dependent transcription by sequestering negative regulators without binding DNA or occupying a specific site via another factor. This mechanism is also independent of DNA binding. At the moment, we have evidence supporting these models without excluding one or the other. It is also possible that GFI1B is present at sites prior

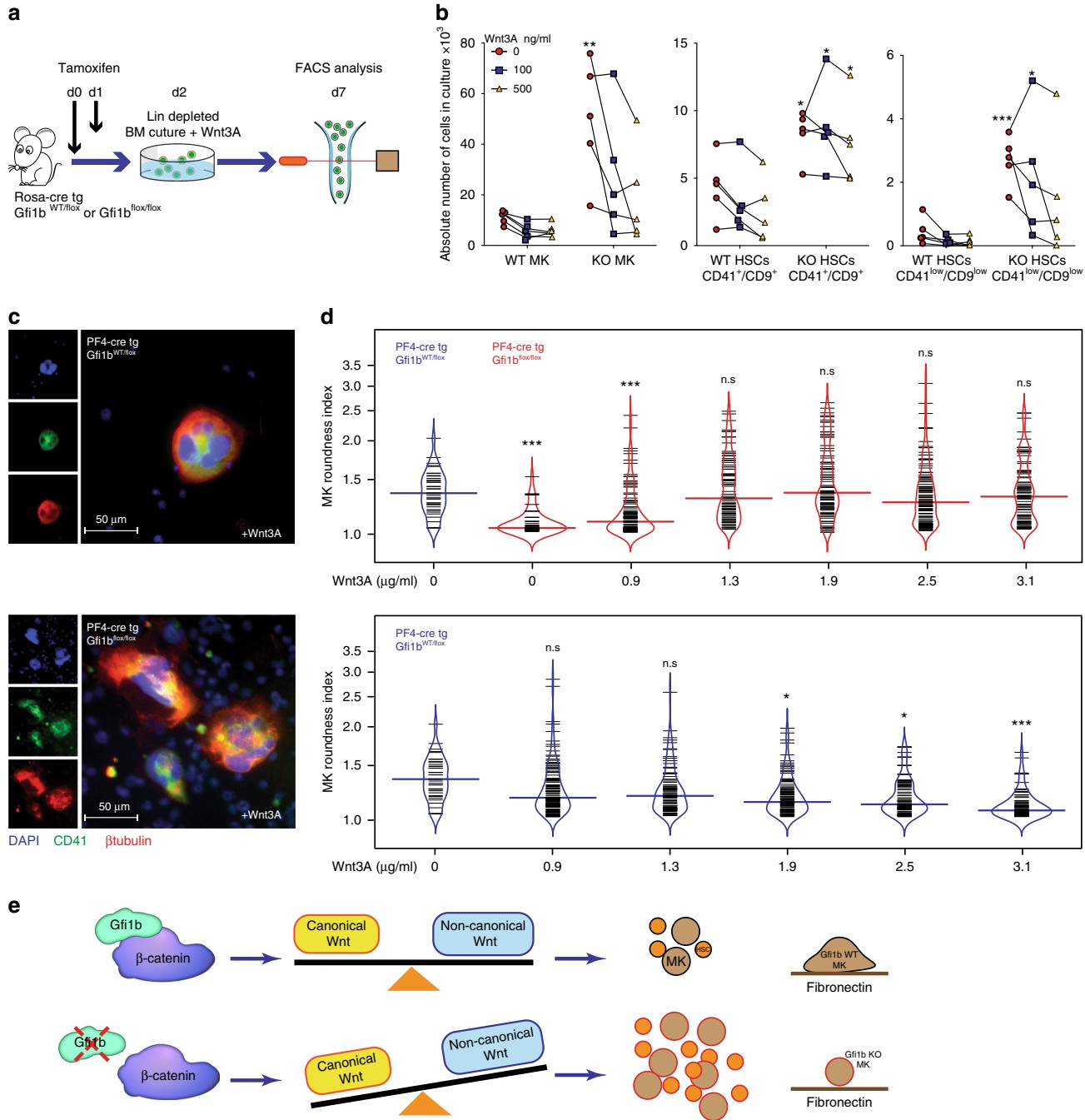

**Fig. 9** Up-regulation of canonical Wnt signaling rescues Gfi1b-deficient phenotypes in HSCs and MKs. **a**, **b** Lineage depleted Gfi1b WT/KO BM cells were taken into culture under the indicated concentrations of Wnt3A. MKs and HSCs CD41$^{low}$/CD9$^{low}$ were quantified by FACS at day seven post Tamoxifen injection, $n = 5$ GFI1b WT and five GFI1B KO mice. **c** Gfi1b KO MKs spreading on fibronectin-coated matrices is rescued by Wnt3A treatment. **d** MK roundness index plotted as bean plot. Upper part: spreading of Gfi1b KO MKs on Fibronectin coated matrices is rescued by Wnt3A treatment at increasing concentrations. Lower part: spreading of WT MKs on Fibronectin coated matrices is abrogated by Wnt3A treatment at increasing concentrations. (*$p < 0.05$, **$p < 0.001$, ***$p < 0.0001$ on a Mann–Whitney U-test, colored horizontal line shows median, $n$ values for upper part are, in order from left to right: 39, 46, 95, 83, 91, 111, and 87. $n$ values for lower part are, in order from left to right: 39, 120, 89, 108, 86, and 81). Each horizontal short line represents one individual cell. **e** Proposed model: Gfi1b interacts with β-catenin and regulates Wnt signaling in HSCs and MKs. Deletion of Gfi1b alters the balance between canonical and non-canonical Wnt signaling leading to expansion of HSCs and MKs and impaired MKs spreading on Fibronectin coated matrix

to the recruitment of β-catenin as a form of priming, and that once β-catenin is recruited to these sites, it forms a new complex with GFI1B that is responsible for the regulation of the gene. The fact that recruitment of β-catenin to many of these sites is reduced in the context of GFI1B knockdown supports this hypothesis.

While Gfi1b deletion leads to both HSC and MKs expansion and, upon Wnt3a treatment, a significant decrease in expansion in both cell populations, there are differences in the expression of Wnt target genes between these cell populations (Fig. 6d). This may be due to different transcription factors and other gene regulator within the two different cell populations. For example, self-replicating and

pluripotent HSCs distinctly contain Gfi1 and PU.1, while the more differentiated MKs contain Gata1 and Fli1 which are not prominent in HSCs[52,53]. These transcription factors could distinctly influence Gfi1b's genomic target regions within these different cells populations and lead to expressional differences of Wnt target genes, further highlighting the need for future research.

Our finding that treatment of Gfi1b-deficient HSCs and MKs with Wnt3A treatment can rescue both the expansion of HSCs and MKs, as well as the spreading of Gfi1b-deficient MKs strongly supports a role of Gfi1b as a regulator of the Wnt/β-catenin pathway. This may be possible through β-catenin recruitment to target loci that do not depend on Gfi1b, such as AXIN2 and SP5, but which nevertheless are targeted by the tri-partite complex. Other transcription factors like, Gata1, Fli1, TCF1 or LEF1 could potentially be influencing factors of β-catenin recruitment in these cases. Further evidence comes from our experiments using retroviral transduction of active β-catenin into bone marrow cells of Gf1b- deficient mice, which led to a significant reduction of HSCs and MKs cell numbers in mice transplanted with transduced BM cells.

Our findings describe an unexpected function for GFI1B in modulating Wnt/β-catenin signaling, which differs from its traditional activity as a transcriptional repressor, since mutants that lack direct DNA binding still activate TCF dependent transcription and the genomic sequences at sites of co-occupation of Gfi1b/β-catenin and LSD1 lack the classical Gfi1/Gfi1b DNA consensus motif, suggesting that GFI1B may act as part of yet undescribed regulatory complexes. This new function of Gfi1b as a regulator of the Wnt/β-catenin pathway appears essential for HSCs and MKs to maintain their cellularity and for MKs to respond to integrin ligands properly (Fig. 9e). Although further investigation into the precise mechanism of this novel role is required, it is plausible that the level of Gfi1b expression is critical not only for the fine-tuning of the expression of Wnt/β-catenin target genes, but also for the balance between canonical and non-canonical Wnt signaling in these cells to maintain their functionality.

## Methods

**Mice**. Gfi1b$^{fl/fl}$ mice[9] were crossed with Rosa-CreERT2, and Axin2$^{+/LacZ}$ (Jackson, B6.129P2-Axin2tm1Wbm/J) or TCF/Lef:H2B-GFP (Jackson, 61Hadj/J) transgene reporter line. PF4-cre mice were used to delete Gfi1b in MKs for immuno-fluorescence microscopy analysis. Age and sex-matched mice were used in all experiments. For BM transplantation C57B6 CD45.1 mice were used. All mice were housed under SPF conditions and the IRCM Institutional Review Board approved all animal protocols and experimental procedures were performed in compliance with IRCM and CCAC (Canadian Council of Animal Care) guidelines.

**Treatment**. Suboptimal doses of tamoxifen were injected at 100 mg per kg at day 0 followed by 50 mg per kg the day after, and mice were analyzed at day 10.

**Flow cytometry analysis, sorting of HSCs and MKs**. Hematopoietic cells were analyzed with LSR, or LSR Fortessa flow cytometers (BD Biosciences, Mountain View, CA) and analyzed using BD FACS Diva software (BD Biosciences) or FlowJo (for histogram overlays; Tree Star). For cell sorting, lineage negative BM cells were first depleted using mouse lineage cell depletion kit (Miltenyi Biotec) then applied to five-laser FACSAria II sorter (BD Biosciences).

**Cell culture**. K562 (ATCC CRL-3344), HEL (ATCC TIB-180) cells were maintained in RPMI media (Multicell) supplemented with 10% Bovine Growth Serum (RMBIO Fetalgro) and 100 IU Penicillin and 100 μg/ml Streptomycin (Multicell). HEK-293 (ATCC CRL-1573) and U2OS (ATCC HTB-96) cells were maintained DMEM media (Multicell) with above mentioned supplements. We verified that none of the cell lines used in this study were found in the Register of Misidentified Cell Lines maintained by the International Cell Line Authentication Committee.

**Antibodies**. The following  antibodies were used for western blots analysis at 1/1000 dilution: β-catenin (cell signaling 9566 and 4176), TLE1 (sc-9121), TLE3 (sc-13374), Pontin52 (ab133513), CHD8 (cell signaling 7656), LSD1 (sc-67272 and cell signaling 4064), Gfi1b (ARP30094 and sc-28356), Tubulin (cell signaling 9099), G6PD (ab993), HDAC1 (ab702A). Uncropped and unprocessed scans of the western blots are provided in Supplementary Data 3.

**Immunofluorescence of Gfi1b WT/KO MKs**. Lineage depleted bone marrow from PF4-Cre Gfi1b-flox/flox and Gfi1b-wt/flox mice were cultured in Stemspan/2.6% FBS supplemented with 1% L-Glutamine and SCF (20 ng per ml). TPO (50 ng per ml) was added to fresh medium at day 2 and cells were cultured for 4 more days. MKs were enriched on a BSA gradient and plated on fibronectin (500 μg per ml; Life Technology) coated 12-μ-chamber slide (ibidi). Cells were incubated at 37 °C, 5% CO$_2$ for 3–6 h with or without Wnt3A (R&D systems) allowing megakaryocytes to attach and spread. After incubation cells were fixed with 4% formalin, permeabilized with 0.1% Triton-X100 in PBS, and blocked with FcBlock (1:500; BD Biosciences). Cells were then labeled with FITC-CD41 (BD Biosciences), AF555-β-tubulin (Cell Signaling) and covered with Vector Shield containing DAPI (Vector Laboratories). Immunofluorescence imaging of the slides was done using a DCX-950P DP72 camera (Sony) mounted on a Leitz DMRB microscope (Leica). Images were analyzed using ImageJ v1.46r (NIH).

**BioID-MS data analysis**. The BioID-MS data were analyzed using the ProHits software[54,55]. The Proteowizard4 tool was used to convert RAW files to.mzXML files. Peptide search and identification were performed by using Human RefSeq version 57 and the iProphet tool integrated in ProHits[56].

**GO term enrichment analysis**. Functional annotation of GO terms was analyzed by using the g:Profiler tool[57]. Biological process or molecular function of prey proteins identified in GFI1B-BirA*-Flag BioID/MS or β-catenin-BirA*-Flag BioID/MS with the indicated constructs are shown in heat map analyses. Reviewed UniProtKB entries of the prey proteins analyzed in Significance Analysis of INTeractome (SAINT) results generated in ProHits were entered in the Query field on g:Profiler for GO term analysis[54,55,58]. Contaminant proteins identified in GFI1B-Flag AP/MS were filtered by using the Contaminant Repository for Affinity Purification (CRAPome)[59] repository prior to the GO term analysis on g:Profiler. The enrichment score of GO terms is shown as the −log10 of corrected P values.

**Dot plot analysis**. SAINT output files of GFI1B-BirA*-Flag BioID/MS or β-catenin-BirA*-Flag BioID/MS data analyzed in ProHits were submitted in ProHits-viz to perform dot plot analyses[60].

**TOP/FOP Flash reporter assays**. 293T Cells were transfected with TOP/FOP Flash reporters, β-galactosidase, and effector plasmids using Lipofectamine® (ThermoFisher) and luciferase activity was normalized by β-galactosidase activity[61].

**Quantification of intracellular β-galactosidase activity**. Lineage negative BM cells were loaded with 2 mM Fluorescein di-β-D-galactopyranoside (FDG) substrate (AAT Bioquest) by hypotonic shock at 37 °C for 5 min, prior to cell surface antibody staining. β-galactosidase reaction was stopped with 1 mM Phenylethyl β-D-thiogalactopyranoside (PETG, Sigma).

**Gene expression profiling by RNA-seq analysis**. Bone marrow from 2 tibiae, 2 femora and 2 humeri (from 5 Rosa-Cre Gfi1b$^{fl/fl}$ and 10 Rosa-Cre Gfi1b$^{wt/fl}$ mice treated with tamoxifen 2 weeks prior the experiment) was harvested in PBS/2.5% FBS and pooled prior to lineage negative depletion using autoMACS Pro separator (Miltenyi Biotec). Cells were incubated with a lineage antibody cocktail (B220, CD3, CD4, CD8, Gr1, CD11b, NK1.1, Il7R, CD19) and then labeled with PE/Cy5-streptavidin, PE-anti-CD41, AF700- anti-CD9, APC- anti-CD150, BV421-anti-cKit, BV510-anti-CD48 and PE/Cy7-anti-Sca1 antibodies. MKs (Lin$^-$ cKit$^+$ CD41$^+$ CD9$^+$), HSCs (Lin$^-$ cKit$^+$ Sca1$^+$ Cd48$^-$ CD150$^+$), and HSCs CD41$^{low}$ CD9$^{low}$ were sorted on FACSAria II sorter (BD Biosciences). RNA was extracted using MagMax-96 Total RNA Isolation kit (Ambion) and quality-checked with RNA 6000 Pico kit (Agilent). RNA-seq libraries were prepared from the RNA extracts using the Illumina TruSeq Stranded mRNA Kit according to the manufacturer's instructions, and sequenced using the TruSeq PE Clusterkit v3-cBot-HS on an Illumina HiSEq 2000 system. Sequencing reads were aligned to the mm10 genome using Tophat v2.0.10[62]. Reads were processed with Samtools[63] and then mapped to Ensembl transcripts using HTSeq[64]. Differential expression was tested using the DESeq R package[65](R Core Team 2015, http://www.r-project.org/). A genome coverage file was generated and scaled to RPM using Bedtools[66]. RNA-seq data are available under accession number GSE71310 (MKs) and GSE85737 (HSCs). Total read numbers and aligned read numbers for each experiment is shown in Supplementary Data 1.

**Functional analysis**. The enrichment of selected biological functions of interest (Supplementary Table 1) was also analyzed using the GSEA tool[67]. Normalized read counts for Ensembl genes from HTSeq were used and enrichment calculated using 1000 Gene Set permutations. Unsupervised clustering analysis was done using web tool ClustVis (https://biit.cs.ut.ee/clustvis/).

**Consensus motif analysis**. Motif scanning was performed using the AME tool from the MEME Suite[68] using the JASPAR CORE 2016 database.

**Chromatin immuno-precipitation (ChIP)**. GFI1B, LSD1, β-catenin and histone modification ChIPs were performed on $1–20 \times 10^6$ K562 cells treated or untreated in culture with 100 ng per ml rhWnt3a (5036-WN; R&D Systems) or CHIR99021 (SML1046;Sigma) for 4 or 8 h, respectively. The cells were cross-linked with 1.5 mM EGS for 20 min and 1% formaldehyde for 8 min before quenching with 125 mM glycine. Cells were lysed in lysis buffer and sonicated using a Covaris E220 to generate 200–600 bp fragments[69]. Samples were immuno-precipitated with 2–5 µg of either anti- GFI1B (ARP30094_P050; Aviva Systems Biology), anti-LSD1 (ab17721;Abcam), anti-β-catenin antibody (71–2700; ThermoFisher), anti-H3K4me1 (ab8895, Abcam), anti-H3K4me2 (ab11946; Abcam), anti-H3K4me3 (ab8580; Abcam) or anti-H3K9me2 (ab1220; Abcam). Libraries were generated according to Illumina's instructions. Libraries were sequenced on the Illumina Hi-seq 2000 following the manufacturer's protocols to obtain 50 bp paired end reads. ChIP-seq results for Gfi1b binding in HPC7 cells[38] and CTCF in ESCs[46] were obtained from the laboratory website of Dr. Goettgens (http://hscl.cimr.cam.ac.uk/). Results for β-catenin[39] and LSD1[40] binding in ESCs were obtained from GEO accessions GSE43597 and GSE22557, respectively. External datasets were obtained in the form of.bed files of peaks and.wig visualization tracks, aligned to the mm9 build, with the exception of LSD1, which only included the.bed peak file.

**Annotation databases used**. For gene promoters, we used the Ensembl Genes 92 database, dataset GRCh38.p12. (https://useast.ensembl.org/index.html) For enhancer regions, we used the Fantom5 human_permissive_enhancers_phase_1_and_2 enhancers (February 2015) dataset (http://fantom.gsc.riken.jp).

**Megakaryocytes in vitro culture and roundness index**. Primary megakaryocytic cultures were prepared by flushing bone marrow into DMEM/10% FBS and suspended in 1x ACK red blood cell lysis buffer (0.15 M NH$_4$Cl, 10 mM KHCO$_3$, 1 mM EDTA). Cells were then labeled with a biotinylated lineage antibody cocktail (B220, Gr1, CD11b, CD16/32) and anti-biotin magnetic beads and separated on an AutoMACS. The negative fraction was washed with DMEM/10% FBS and suspended in StemSpan SFEM (StemCell Technologies) that contained 2.6% FBS, 1% L-Glutamine and SCF (20 ng per ml). Cells were then cultured two days at 37 °C and 5% CO$_2$. The media was then replaced with fresh StemSpan SFEM that contained 2.6% FBS, 1% L-Glutamine, SCF (20 ng per ml) and TPO (50 ng per ml) and cells were cultured for 4 more days. On the 7th day, mature megakaryocytes were enriched on a BSA gradient and plated in fibronectin coated 12-well µ-Chamber glass slides (ibidi). Cells were then allowed to attach and to spread for 3–8 h at 37 °C under 5% CO$_2$. To quantify the spreading of MKs on coated slides, we compared the periphery (P) of each cell to the circumference of a perfect circle ($2\pi r$) having the same area as this cell and calculated a roundness index (RI) resulting in a numerical value that increases as the complexity of a spreading cell increases, starting at RI = 1 for a perfectly round cell[11].

**Statistical analysis**. The paired Student $t$-test was chosen for analyzing the differences in the intracellular β-galactosidase activity in MKs and HSCs. Welch corrected Student's $t$ tests were used to calculate the statistical significance of results from luciferase reporter assays. Overlap between binding sites was calculated using Fisher's exact test. All $p$-values were calculated two-sided, and values of $p < 0.05$ were considered statistically significant. Statistical analysis was done with GraphPad Prism software (GraphPad software, La Jolla, CA, USA). The sample size of data points for each assay is shown in Supplementary Data 2.

**Reporting summary**. Further information on experimental design is available in the Nature Research Reporting Summary linked to this article.

## Data availability
The raw proteomics data, which are presented in Figs. 2a, b, and 3d, is publicly available on massIVE [https://massive.ucsd.edu/ProteoSAFe] under the following accession number: MSV000083125. The raw ChIP-seq and RNA-seq data, which is presented in Figs. 6–8 and Supplementary Figures 3 and 7, have been uploaded to the GEO Datasets repository [https://www.ncbi.nlm.nih.gov/gds] and is available under the following accession numbers: GSE71310, GSE85737 and GSE117944.
Previously published ChIP-seq data, which is presented in Supplementary Figs 4 and 5, is available under the following accession numbers: GSE43597, GSE22557.

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

## Acknowledgements

We are indebted to Mathieu Lapointe for technical assistance, Marie-Claude Lavallée and Jo-Anny Bisson for excellent animal care, Eric Massicotte and Julie Lord for FACS and cell sorting, Virginie Calderon for assistance with bioinformatics analysis. We thank Genome Quebec for performing HT sequencing. Tarik Möröy holds a Canada Research Chair (Tier 1) and grants from the CIHR (MOP-84238, MOP-94846, FDN-148372) and from the Canadian Hemophilia Society. Charles Vadnais was supported by fellowships from the Fonds de recherche Quebec – Santé (FRQS) and from the CIHR. Halil Bagci was supported by a doctoral training award from FRQS (#33603) and NSERC Discovery Grant (RGPIN-2016-04808 to Jean-François Côté). Jean-François Côté holds the TRANSAT chair in breast cancer research. Tarik Möröy is supported by a Canada Research Chair (Tier 1) and by grants from CIHR (MOP-84238, MOP-94846, FDN-148372) and the Canadian Hemophilia Society.

## Author contributions

P.S.: Concept and design, Collection of data, data analysis, manuscript writing and final approval, A.H.: Concept and design, Collection of data, data analysis, manuscript writing. C.V.: Data analysis, manuscript writing, N.B.: Collection of data, data analysis, H.B.: Collection of data, data analysis, R.C.: Concept and design, collection of data, H.B.: Data analysis, F.J.T.S.: Concept and writing, reagents, J-F.C.: Concept and writing, reagents. T. M.: Concept and design, data analysis and supervision, manuscript writing, final approval, provision of funds.

## Additional information

**Competing interests:** The authors declare no competing interests.

