## [Peer Review File · Nature Communications]

Reviewers' comments:

Reviewer #1 (Remarks to the Author):

Deletion of the transcriptional repressor Gfi1b leads to expansion of both HSCs and MKs. Gfi1b deleted HSCs remain functional, however Gfi1b deleted MKs are no longer capable of producing platelets and responding to integrin receptor stimulation. How Gfi1b controls these processes remains unclear. Here the authors provide data for a link between Gfi1b and the wnt/b-catenin pathway. At the cellular level they show that loss of Gfi1b leads to decreased levels of Wnt signaling in HSCs and MKs, which can be reversed by treatment with Wnt3a. At the molecular level they show that Gfi1b can enhance transcription of TCF/LEF dependent promoters by recruiting LSD1 to b-catenin containing complexes. The major issue with the data is that the functional link between Gfi1b and wnt signaling is shown in HSCs and MKs, whereas all the molecular analyses have been performed in 293T cells. Of course most of these experiments are difficult to perform in hematopoietic subpopulations due to limited cell numbers. However there are hematopoietic derived cell lines available one could use. In addition, the data still do not provide convincing evidence for the differential outcome of deletion of Gfi1b in HSCs versus MKs.

Minor comments:

- there is no reference in the text to fig S1b
- Fig2a: where there also other binding partners identified than the ones listed, belonging to histon modifying enzymes and wnt pathway?
- Fig 2c: with the mutant lacking the WRD domain there is still some binding of TLE1, how do the authors explain this?
- Fig 2e: did the authors look at binding of b-catenin as well in this experiment?
- Fig 4c: are the lines connecting the FDG MFI of wt and KO cells in the same mouse? There is a lot of variation in the data. Some lines (mice?) do not show any difference between wt and KO, especially in HSCs. What is the explanation here?
- And are these widespread data points of wt in Fig 4e again due to variation in mice like in 4c? The data in 4c and 4e would suggest high variation or maybe even two subsets of HSCs responding differently to Gfi1b KO?
- The functional outcome of Gfi1b KO in HSCs and MKs seems different with Gfi1b KO functionally affecting MKs but not HSCs: does the RNAseq data in Fig 5 reveal differences in the mechanism between the 2 cell types?
- Now that the potential mechanism of Gfi1b is more clear from the data in the 293T cells it would be interesting to perform for example CHIPseq as in Fig 7 with MKs and/or HSCs to see whether what was found in 293T cells holds true for HSCs and MKs

Reviewer #2 (Remarks to the Author):

This manuscript describes data in support of a tripartite β -catenin/Gfi1b/LSD1 complex that regulates expression of Wnt/ β -catenin target genes. It is argued that Gfi1b is in complex with β -catenin and other β -catenin co-factors (incl. Pontin52, CHD8, TLE3 and CtBP1). Furthermore, evidence is presented that Gfi1b is required for the activation of β -catenin/TCF dependent transcription and that TCF-dependent transcription by Gfi1b is increased after treatment with Wnt3a and requires the interaction between Gfi1b and LSD1. Expression of Wnt/ β -catenin target genes is decreased in Gfi1b-deficient cells, and some target genes are co-occupied by Gfi1b, β -catenin and LSD1 at promoter sites. Finally, Wnt3a treatment of Gfi1b deficient cells restores normal cellularity and MK's ability to spread on integrin. Overall these data suggest that Gfi1b acts as a co-factor that acts to augment Wnt target

gene expression, perhaps by sensitizing the cellular response to endogenous Wnt signals. There are several major and minor points that need to be addressed.

Major points (not in order of importance):

Figure 2a: LC/MS2 analysis detects many proteins, many of which are not necessarily associated with the target protein. The authors should provide additional analysis on the enrichment of Wnt pathway components in this Gfi1b immunoprecipitate. Use of GO or GSEA may provide some measure of enrichment.

Figure 2d and e: the co-immunoprecipitation of b-catenin (and to a lesser extent Pontin 52) with Gfi1b appears to be non-specific because substantial amounts co-IP with a presumably non-specific IgG. If the authors wish to argue that there is an enrichment of these proteins in these Gfi1b IPs then they should increase the stringency of washes to reduce the amount of background binding. As shown, these results are unconvincing.

Figure 2f: the increased amount of LSD1 in a b-catenin IP in the presence of overexpressed WT Gfi1b versus mutant Gfi1b is unimpressive, given that this mutant Gfi1b protein shows no binding to LSD1, as shown in panel c. Or are the authors arguing that b-catenin binds LSD1 independently of Gfi1b?

Figure 3g and h: Gfi1b counteracts the inhibitory effect of TLE1 and CtBP1 on b-catenin reporter activity. Is this due to displacement of TLE1 or CtBP1 from the complex or to direct binding of Gfi1b to TLE1 and CtBP1 as suggested by co-IP experiments shown in Figure 2c? Does overexpressed Gfi1b act to titrate out these repressors, thereby elevating Wnt reporter activity? This alternative model should be addressed.

Figure 6: the ChIP analysis is quite superficial and should be expanded upon. Does the Gfi1b ChIP enrich the Gfi1b binding motif? A motif enrichment analysis will reveal overrepresented motifs, and ideally the known Gfi1b motif (AAATCTCTGCA) is observed. Furthermore, given the presumed close association of Gfi1b and TCF-binding sites, does such an analysis identify the known TCF binding site? Is there perhaps a subset of genes associated with Gfi1b peaks that have a TCF binding site? Such sites may be locations where Gfi1b binds to TCF/b-catenin rather than to DNA via its own binding motif. To expand on this point: a prediction of the data shown in Figure 3i and j is that expression of target Wnt genes with a Gfi1b binding site near the TCF/LEF binding site will be inhibited by Gfi1b overexpression. Further, expression of such Wnt target genes should increase upon Gfi1b KD. In contrast, Wnt target genes with a Gfi1b ChIP peak overlapping the TCF/b-catenin peak should be increased upon Gfi1b overexpression. Is this the case?

Figure 7d: the increase in expression of CCND1 in Gfi1b and LSD1 transfected cells versus LSD1 alone is not impressive and does not support the statement (page 15): "the expression of CCND1 was increased in HEK293 cells upon transfection by Gfi1b or LSD1 and was further enhanced by a combination of both factors (Fig. 7d)."

Figure 7f: is expression of Wnt target genes that are sensitive to Gfi1b KD also affected as expected when Wnt is activated or inhibited? For example, how does treatment of cells with Porcn inhibitors affect expression of these target genes? What about in presence and absence of Gfi1b? And what about in the presence and absence of exogenous Wnt signaling?

Minor points:

Figure 1b and c: these results (increase in HSC and MK numbers in Gfi1b knockouts) were previously

published by the authors; how are these results different and/or new compared to previous study?

Figure 1f and 8c: include scale bar for each panel. For Figure 1f, it appears that images of WT cells are at a higher magnification than images of mutant cells; if this is the case, the authors should adjust the magnification so that they are the same. Also in Figure 1f, large image on left: what are all the small blue dots surrounding the main cell?

Figure 2b: the alignment of the WRD sequence is confusing. Are these all sequences of Gfi1b from different species? Are the amino acid positions 106 to 111 identical for Gfi1b from all shown species? Is the muWRD sequence from mouse Gfi1b? How is the consensus sequence at the top derived? It appears that FSWD is completely conserved yet the consensus lists it as FS/TXX. Please clarify.

Figure 3c-e: are these various mutant versions of Gfi1b expressed at equal levels? An immunoblot showing that each protein is present should be included. Are these transgenes Flag-tagged as indicated in Figure 2b?

Page 14: the following statement is poorly supported and lacks quantitation: "This finding indicates that loss of Gfi1b can lead to both up- and down-regulation of β -catenin target genes in primary hematopoietic cells and changes the balance between canonical and non-canonical Wnt target genes." To what extent is the balance between canonical and non-canonical Wnt signaling altered in Gfi1b deficient cells?

Figure 5b and 6c: references to each gene set should be provided.

Figure 7b: this looks like RNA-seq data, not ChIP-seq as indicated in the figure legend.

Throughout the figures the authors switch between "Gfi1b" (indicating mouse origin) and "GFI1B" (indicating human origin). Is this an intentional distinction? Please clarify.

Abstract: The abstract would benefit from some editing to streamline and clarify what the authors are trying to present. As it reads now, it is rather difficult to follow.

Figure 2a: Table has a typo: Histone.

Figure 3c-h: Why have the authors now switched to LiCl as a Wnt stimulus? LiCl targets GSK3 and is woefully nonspecific. These assays should be performed in the context of Wnt stimulation, to demonstrate that this effect is due to Wnt. Also, much better GSK3 inhibitors are available and could be employed.

Figure 3i-j: All of the other data in this figure is reported as TOP/FOP. Why have the authors now eliminated the FOP normalization? These experiments should be repeated in the context of the FOP reporter with the Gfi1b binding sites for normalization, if we are to compare these assays.

Reviewer #3 (Remarks to the Author):

The manuscript by Shooshtarizadeh et al, identifies physical association of β -catenin and other Wnt pathway components with the transcription factor Gfi1b by immuno-precipitation followed by mass spectrometry. The authors then endeavor to demonstrate Gfi1b mediated activation of Wnt/ β -catenin signaling in HSC and megakaryocyte cellularity and function. They do so by demonstrating the impact of Gfi1b and its co-factor LSD1 on Wnt/ β -catenin transcriptional regulation by promoter driven

reporter assays and by Gfi1b/LSD1 dependent modulation of Wnt target gene expression in hematopoietic and non-hematopoietic cells. Finally, they assert that attenuation of Wnt signaling in the Gfi1b deficient HSCs and MKs, can be partially rescued by external treatment with Wnt3a. Although the Gfi1b-Wnt pathway interaction is an interesting and novel finding and the authors perform several and diverse kinds of follow up experiments to address their combinatorial role in hematopoietic development, the paper does not lead to a mechanistically compelling scenario of the co-operativity between these pathways in this complex and vital process. As discussed below much of the data is circumstantial and a number of unexpected and conflicting observations are not adequately addressed or resolved.

Figure 1: Does deletion of exons 2-4 produce a true null phenotype? Does a truncated protein result from the mutated transcript and if so does it have any residual or dominant negative function?

Figure 2: Extent of Gfi1b and LSD1 precipitation is not indicated in the mass spec data. Reasons for using 293T instead of hematopoietic cells not justified.

2c. β -catenin immunoprecipitates with P2AGfi1b and Gfi1bmWRD ie region of Gfi1b interacting with β -catenin is undefined. 2d. Co-IP of β -catenin in K562 cells only marginally better with Gfi1b antibody relative to IgG control. 2e. HEL-Same for pontin52; β -catenin not shown. 2f-g. Residual LSD1 precipitated with β -catenin in SNAG mutants not explained. Is the association of LSD1 with β -catenin partly Gfi1b independent?

Figure 3: Why is transcriptional activation shown as a ratio of TOP/FOP luciferase activity and not relative to the TOP vector alone (except in i and j) is unclear.

3c-e. Why are different WRD mutants used in reporter assays relative to those in the IP experiments? Also if the WRD mutant continues to bind β -catenin, TLE1/3 and CtBP and is as effective as wt Gfi1b in stimulating TCF mediated transcription, then what is its significance? What Gfi1bmutGBS stands for is not explained in the text or figure legend. 3f. Effect of P2AGfi1b/LSD1 combination on TCF reporter activity would also be more informative in demonstrating the physical basis of Gfi1b/LSD1 transcriptional synergy.

Figure 4c. Shift in intensity of lacZ expression (as a measure of TCF/LEF driven transcription) in gfi1b deficient versus wt cells would be more obvious if their profiles were superimposed rather than of being shown side by side relative to axin+/+ profiles which are lacZ negative.

How many mice were used for these analyses and statistical significance are not addressed?

No positive or negative controls are shown to establish the specificity versus randomness of the results.

Figure 5: Reasons or mechanisms for up- versus down-regulation of Wnt target genes were not adequately addressed upon Gfi1b deletion given that the latter is a putative positive regulator of β -catenin transcription.

Figure 6: This figure is rather artifactual and difficult to interpret as it is a compilation of data from different sources and cells (HPC7 versus ES). In order to determine common targets of Gfi1b and β -catenin in HSCs or megakaryocytes, ChIP experiments need to be performed in parallel for both proteins in the same cells under identical conditions. As such figure 6A shows only minute overlap between Gfi1b and β -catenin genomic targets. ~10% of Gfi1b targets are common to β -catenin and <4% of β -catenin targets are bound by Gfi1b. Is this extent of overlap statistically or biologically meaningful or even specific.

Figure 7: Unclear what proteins are being ChIPed in (a) and (b) or what cells were used in (a). ChIP results in 293 cells over-expressing β -catenin or Gfi1b may produce artifactual results. Why was enrichment of endogenous proteins at these loci in hematopoietic cells not determined instead? Unclear why acetylated H3K9 levels are being monitored in (g) instead of H3K4/K9 methylation levels.

Figure 8: Partial "rescue" of Gfi1b ko phenotype by β -catenin overexpression does not necessarily imply a common pathway but may be due to compensation from distinct and parallel pathways. Since the mechanism by which Gfi1b and LSD1 mediate transcriptional repression are well established, their anomalous behavior in "activating" β -catenin mediated transcription needs to be more

substantially addressed. Particularly the following -

1. Although these actions are proposed to be independent of the DNA binding activity of Gfi1b, the authors do not directly demonstrate recruitment of Gfi1b/LSD1 to Wnt targets by β -catenin or related factors ie in β -catenin deficient or depleted cells.
2. The authors also do not attempt to explain the mechanism of Gfi1b mediated enhancement versus reduction of β -catenin enrichment on subsets of target promoters.
3. Since the SNAG domain of Gfi1b recruits LSD1 it is unclear why is the latter absent on certain β -catenin/Gfi1b targets and present on others.
4. Does the presence or absence of LSD1 impact the chromatin status of the loci particularly H3K4 and H3K9 methylation since these two residues are known to be demethylated by LSD1.
5. Since Gfi1b also interacts with HDACs (previously shown and in Fig 2), does its presence on Wnt targets lead to HDAC recruitment.

Additionally other questions that need to be addressed include-

1. Do Wnt signaling or β -catenin association impact DNA binding dependent transcriptional effects of Gfi1b?
2. A coherent model that explains the data and ties in Wnt signaling with Gfi1b/LSD1 in HSC and megakaryocyte development and differentiation.

We thank the Editor and reviewers for their insightful comments and patience. We have spent a significant amount of time and effort working to obtain additional RNA-seq data and in particular new Gfi1b, LSD1 and beta-catenin ChIP-seqs in K562 cells to detect genome wide occupation of these factors at their endogenous expression levels. The three first authors of this study have contributed to a very similar extent to this major effort and to reflect their role in this study, I propose to list them as equal contributors in the author list.

The new experiments have been done in order to reply to requests from reviewers; below is a point-by-point response to each comment with our response in blue font and the original comments in black. A few figures were excluded or rearranged and new figures were added to the revised manuscript. We mention the new Figure number in each answer accordingly.

Reviewer #1 (Remarks to the Author):

Deletion of the transcriptional repressor Gfi1b leads to expansion of both HSCs and MKs. Gfi1b deleted HSCs remain functional, however Gfi1b deleted MKs are no longer capable of producing platelets and responding to integrin receptor stimulation. How Gfi1b controls these processes remains unclear.

Here the authors provide data for a link between Gfi1b and the wnt/b-catenin pathway. At the cellular level they show that loss of Gfi1b leads to decreased levels of Wnt signaling in HSCs and MKs, which can be reversed by treatment with Wnt3a. At the molecular level they show that Gfi1b can enhance transcription of TCF/LEF dependent promoters by recruiting LSD1 to b-catenin containing complexes. The major issue with the data is that the functional link between Gfi1b and wnt signaling is shown in HSCs and MKs, whereas all the molecular analyses have been performed in 293T cells. Of course most of these experiments are difficult to perform in hematopoietic subpopulations due to limited cell numbers. However there are hematopoietic derived cell lines available one could use. In addition, the data still do not provide convincing evidence for the differential outcome of deletion of Gfi1b in HSCs versus MKs.

We feel that we have provided functional experiments in hematopoietic cells since our results with two different Wnt reporter mouse models (Fig. 4) show that Wnt/b-catenin signalling is altered in Gfi1b KO MKs and HSCs. As suggested by both the reviewers and the editor, we performed Gfi1b, b-catenin and LSD1 ChIP-seq analysis at their endogenous expression levels in a more relevant haematopoietic cell line in order to confirm our preliminary observation using published ChIP-seq data from embryonic stem cell lines. We have chosen the haematopoietic cell line K562, since Gfi1b is expressed here and both LSD1 and b-catenin are present in detectable levels. We were able to generate Ch-IP seq data for Gfi1, LSD1 and b-catenin. This is to our knowledge the first b-catenin Ch-IP seq experiment at endogenous expression levels or without a tagged version. These new results were included in Figure 7 and in a supplementary Figure 6. The previously used ChIP-seq data from ES cells and from HPC7 cells were removed from the main Figures, but are still mentioned in the text and are shown in supplementary Figures 3 and 4a-c.

We have published on the differential outcome of the Gfi1b deletion for HSCs and MKs previously and we explain this again but also have to refer the reviewer to our papers from Beauchemin et al., 2017 and Khandanpour et al., 2010 (ref 9 and 25 in the manuscript). Briefly, Gfi1b KO affects megakaryocyte proliferation and ploidy, abrogates their responsiveness towards integrin receptors. Gfi1b MKs are small and cannot spread on surfaces and are unable to reorganize their cytoskeleton; they cannot form pro-platelets. Gfi1b Ko HSCs have no apparent changes in morphology; they can still be transplanted and give rise to all blood lineages, but they expand in numbers, expand into the blood stream (see also Figure. 1) and are no longer quiescent and contain elevated levels of ROS. We have suggested that Gfi1b regulates HSCs cellularity and dormancy.

Minor comments:

- there is no reference in the text to fig S1b

We have corrected this and now refer to this Figure on page 13, first paragraph.

- Fig2a: where there also other binding partners identified than the ones listed, belonging to histon modifying enzymes and wnt pathway?

Besides the mentioned proteins, we did not identify any other known Gfi1b interacting proteins (such as SUV39H1). As suggested by Reviewer 2 we used a GO-term enrichment analysis (see Figure 2b), which revealed that precipitated proteins were enriched within Wnt regulatory pathways. Interestingly, with this, more Wnt regulatory proteins were found (such as: Spindlin1, UBR5, CDC73, CCAR2 and BRG1) and we added them to the list. To confirm these data we also performed a new BioID experiment (see Figure 2d-f), which also revealed that Gfi1b interacts with proteins and regulators of the Wnt pathway, notably TLE1 and TLE3 and others. We have also added more detail on the mass spectrometry analysis such as the spectra and peptide coverage of the identified proteins.

- Fig 2c: with the mutant lacking the WRD domain there is still some binding of TLE1, how do the authors explain this?

This is now Figure 2h in the revised MS. As opposed to TLE3, the association of TLE1 with Gfi1b was not dependent on the WRD domain (although the association was weaker compared to wt Gfi1b). It is possible that TLE1 associates with other parts of Gfi1b rather than WRD domain or both Gfi1b and TLE1 interact via a third partner.

- Fig 2e: did the authors look at binding of b-catenin as well in this experiment?

This is now Fig 2k in the revised MS. We did, but were unable to detect b-catenin in the blot due to its low level of expression in this cell line.

- Fig 4c: are the lines connecting the FDG MFI of wt and KO cells in the same mouse? There is a lot of variation in the data. Some lines (mice?) do not show any difference between wt and KO, especially in HSCs. What is the explanation here?

Each line shows a pair of mice (WT and KO, sex/age matched, injected with Tamoxifen at the same time) analyzed at the same day. This type of “paired” analysis was necessary due to the variation from one experiment to the other regarding FDG staining and animals. The lack of difference between WT and KO in some pairs can be explained by the variation in Cre-deletion efficiency following Tamoxifen injection.

- And are these widespread data points of wt in Fig 4e again due to variation in mice like in 4c? The data in 4c and 4e would suggest high variation or maybe even two subsets of HSCs responding differently to Gfi1b KO?

Again this can be explained in part by the variation in Cre-deletion efficiency following Tamoxifen injection.

However, as mentioned by the reviewer, we can also consider the possibility that two or more subsets of HSCs with different levels of Wnt/b-catenin signalling alteration following Gfi1b deletion exist.

- The functional outcome of Gfi1b KO in HSCs and MKs seems different with Gfi1b KO functionally affecting MKs but not HSCs: does the RNAseq data in Fig 5 reveal differences in the mechanism between the 2 cell types?

For the functional outcome of Gfi1b Ko in HSCs and MKs see our answer under “point 2” above. Genes differentially expressed between Gfi1b wt and KO HSCs and MKs are depicted in Figure 5 d – these genes had been selected because they are co-occupied by Gfi1b and b-catenin. A gene set

enrichment analysis of RNA seq data from Mks from PF4Cre Gfi1b flox/flox mice was published in our paper Beauchemin et al., 2017 and revealed pathways involving actin cytoskeleton and microtubule growth. RNA expression profiles of Gfi1b wt and Ko HSCs were published in our paper by Khandanpour et al., 2010 and demonstrated altered GSEA profiles for cell adhesion molecules in agreement with our hypothesis that HSCs that lack Gfi1b may be mobilized from the bone marrow to the bloodstream.

- Now that the potential mechanism of Gfi1b is more clear from the data in the 293T cells it would be interesting to perform for example CHIPseq as in Fig 7 with MKs and/or HSCs to see whether what was found in 293T cells holds true for HSCs and MKs

Due to technical difficulties we were not able to do a ChIP-seq analysis in primary MKs or HSCs. However, we performed a new ChIP-seq analysis for Gfi1b, b-catenin and LSD1 in K562 cells (a more relevant haematopoietic cell line) as discussed under point 1 (see above). With these new data the old Chip-PCR results in 293 cells became less relevant and we have removed them from the paper.

Reviewer #2 (Remarks to the Author):

This manuscript describes data in support of a tripartite b-catenin/Gfi1b/LSD1 complex that regulates expression of Wnt/b-catenin target genes. It is argued that Gfi1b is in complex with β -catenin and other b-catenin co-factors (incl. Pontin52, CHD8, TLE3 and CtBP1). Furthermore, evidence is presented that Gfi1b is required for the activation of b-catenin/TCF dependent transcription and that TCF-dependent transcription by Gfi1b is increased after treatment with Wnt3a and requires the interaction between Gfi1b and LSD1. Expression of Wnt/b-catenin target genes is decreased in Gfi1b-deficient cells, and some target genes are co-occupied by Gfi1b, b-catenin and LSD1 at promoter sites. Finally, Wnt3a treatment of Gfi1b deficient cells restores normal cellularity and MK's ability to spread on integrin. Overall these data suggest that Gfi1b acts as a co-factor that acts to augment Wnt target gene expression, perhaps by sensitizing the cellular response to endogenous Wnt signals. There are several major and minor points that need to be addressed.

Major points (not in order of importance):

Figure 2a: LC/MS2 analysis detects many proteins, many of which are not necessarily associated with the target protein. The authors should provide additional analysis on the enrichment of Wnt pathway components in this Gfi1b immunoprecipitate. Use of GO or GSEA may provide some measure of enrichment.

We have performed GO-term enrichment analysis as suggested (see Figure 2b). Interestingly, not only was a Wnt regulatory pathway identified, but more Wnt regulatory proteins were found with this analysis (annotated under "positive regulation of Wnt signalling pathway") and we added them to the list (Spindlin1, UBR5, CDC73, CCAR2 and BRG1). We also performed a new BioID experiment (see Figure 2d-f), which not only confirmed our previous data shown Figure 2a (TLE1, TLE3), but also let us identify new Wnt regulators (e.g. CREBBP) to be associated with Gfi1b. Confirmation of the LC/MS2 and BioID analysis are shown by Co-IP and western blots in figures 2h-n. Furthermore, BioID analysis for b-catenin detected enrichment of GFI1b. (Figure 2o-p)

Figure 2d and e: the co-immunoprecipitation of b-catenin (and to a lesser extent Pontin 52) with Gfi1b appears to be non-specific because substantial amounts co-IP with a presumably non-specific IgG. If the authors wish to argue that there is an enrichment of these proteins in these Gfi1b IPs then

they should increase the stringency of washes to reduce the amount of background binding. As shown, these results are unconvincing.

Figure 2d and e are now Fig. 2j and Fig. 2k in the revised MS. We repeated this experiment using all commercially available antibodies against Pontin52, Gfi1b and b-catenin. We also tried to reduce the amount of background by doing more washes as suggested. Nevertheless we were not able to obtain a cleaner co-IP. But we still believe that there is an enrichment of these proteins over background and we would like to keep these data rather than excluding them from the manuscript, unless specifically requested by the reviewer. Of note we performed a second BioID experiment using b-catenin-BIRA and showed a clear association of GFI1B and b-catenin in this experiment (see Figure 2o, p) confirming our findings from the mass spectrometric analysis of the Flag IP and the co-immune precipitations with transfected cells. Taken together, we feel that these data provide strong, independent lines of evidence that GFI1b interacts with beta-catenin. It is possible that the available antibodies and the levels of endogenous expression of Gfi1b and beta-catenin or the accessibility at the chromatin prevent an optimal co-precipitation.

Figure 2f: the increased amount of LSD1 in a b-catenin IP in the presence of overexpressed WT Gfi1b versus mutant Gfi1b is unimpressive, given that this mutant Gfi1b protein shows no binding to LSD1, as shown in panel c. Or are the authors arguing that b-catenin binds LSD1 independently of Gfi1b?

Figure 2f is now Fig 2l in the revised MS; b-catenin binds independently to LSD1 as was shown previously in several studies such as Song et al. 2015 (Mol Cancer Res. Jun; 13(6): 969–981. Figure 6A) and Yakulov 2013 (Mol Cell Proteomics. Jul; 12(7): 1980–1994. Figure 4A). Fig 2l shows that the presence of Gfi1b enhances the association between b-catenin and LSD1.

Figure 3g and h: Gfi1b counteracts the inhibitory effect of TLE1 and CtBP1 on b-catenin reporter activity. Is this due to displacement of TLE1 or CtBP1 from the complex or to direct binding of Gfi1b to TLE1 and CtBP1 as suggested by co-IP experiments shown in Figure 2c? Does overexpressed Gfi1b act to titrate out these repressors, thereby elevating Wnt reporter activity? This alternative model should be addressed.

Figure 3g and h are now Figure 3h and 3i in the revised MS respectively; Figure 2c is now Figure 2h in the revised MS;

Both possibilities exist; as the reviewer mentions our IP shows that Gfi1b binds to TLE1 and CtBP1. However, we don't have any data to show whether a displacement of TLE1 and CtBP1 takes place following binding to Gfi1b at the promoter. We mention both possibilities in the Discussion. The alternative model was addressed before in the manuscript but we refer to the Figure this time. The text reads: "Alternatively, we show here that Gfi1b interacts with a number of b-catenin inhibitors such as CtBP1 or TLE1. It is therefore conceivable that Gfi1b can also sequester these inhibitors, thereby liberating b-catenin from their inhibitory effect, leading to b-catenin target gene expression. The results from our TOP/FOP reporter assay with Gfi1b in the presence of CtBP1 or TLE1 support this notion (**Fig 3h, i**)"

Figure 6: the ChIP analysis is quite superficial and should be expanded upon. Does the Gfi1b ChIP enrich the Gfi1b binding motif? A motif enrichment analysis will reveal overrepresented motifs, and ideally the known Gfi1b motif (AAATCTCTGCA) is observed. Furthermore, given the presumed close association of Gfi1b and TCF-binding sites, does such an analysis identify the known TCF binding site? Is there perhaps a subset of genes associated with Gfi1b peaks that have a TCF binding site? Such sites may be locations where Gfi1b binds to TCF/b-catenin rather than to DNA via its own binding motif. To expand on this point: a prediction of the data shown in Figure 3i and j is that expression of target Wnt genes with a Gfi1b binding site near the TCF/LEF binding site will be inhibited by Gfi1b overexpression. Further, expression of such Wnt target genes should increase

upon Gfi1b KD. In contrast, Wnt target genes with a Gfi1b ChIP peak overlapping the TCF/b-catenin peak should be increased

upon Gfi1b overexpression. Is this the case?

Figure 6 is now supplementary Figure 3 and supplementary Figure 4a-c in the revised MS.

Analysis of our new ChIP-seq results in K562 cells indeed showed that the GFI1B motif is enriched at locations, where GFI1B binds alone, while several TCF binding motifs (TCF7, TCF7L1, TCF7L2, LEF1) are not enriched (Figure 7c). However, when analysing GFI1b binding sites that overlap with b-catenin binding sites, we found enrichment for the LEF1/TCF consensus motif, but not for GFI1B's own DNA binding motif. These results suggest that GFI1B's binding to these locations (i.e. where it co-localises with b-catenin) may be less reliant on its own motif because of the interaction with b-Catenin and TCF factors. This result also suggests that when in a complex with b-catenin, GFI1B may not require a direct contact with DNA. This notion is supported by our findings with TCF reporter genes with our without GFI1B binding site (see Figure 3j, k).

However, we were not able to find a correlation between the presence of a GFI1B or TCF motif at a given gene promoter and the effect of Gfi1b knockout on the corresponding gene's mRNA expression levels, or with changes in the abundance of histone methylation marks that generally correlate with gene expression or gene priming. We believe this analysis may be inconclusive regarding this specific point, since the ChIP experiments cannot distinguish between direct binding of GFI1b to the chromatin and binding through a protein partner, and since the presence of binding motifs is not always necessary for DNA binding of transcription factors.

Figure 7d: the increase in expression of CCND1 in Gfi1b and LSD1 transfected cells versus LSD1 alone is not impressive and does not support the statement (page 15): "the expression of CCND1 was increased in HEK293 cells upon transfection by Gfi1b or LSD1 and was further enhanced by a combination of both factors (Fig. 7d)."

We removed this Figure from the manuscript and modified the text accordingly.

Figure 7f: is expression of Wnt target genes that are sensitive to Gfi1b KD also affected as expected when Wnt is activated or inhibited? For example, how does treatment of cells with Porcn inhibitors affect expression of these target genes? What about in presence and absence of Gfi1b? And what about in the presence and absence of exogenous Wnt signaling?

Figure 7f is now Figure 5f in the revised MS.

We tried to answer this comment by treating two sets of primary Gfi1b WT/KO MK cells in culture with either Wnt3A or CHIR and then checking the expression of a few genes by PCR or in the RNA-seq data set. The results in both cases (PCR and RNA-seq) were inconclusive probably due to the impact of culture elements such as TPO on Wnt signalling. However, our results presented in Figure 8 suggest that Gfi1b KO MK cells can respond to Wnt3A since their expansion and spreading defect was rescued following Wnt3A treatment.

Minor points:

Figure 1b and c: these results (increase in HSC and MK numbers in Gfi1b knockouts) were previously published by the authors; how are these results different and/or new compared to previous study?

Previously we used an MX-Cre allele for Gfi1b deletion, whereas in this study we use the Rosa-Cre^{ER} deleter. We also used a new gating strategy for our FACS analysis, which is highlighted in Figure 1b, to eliminate MKs before analysing HSCs. We also showed that deletion of Gfi1b in MKs, HSCs CD41^{lo}/CD9^{lo} and HSCs CD41⁺/CD9⁺ leads to the *in vitro* expansion of these cells. These are new

data, but confirm the previously published findings. We wanted to show them here again to precisely define at the beginning of the MS which cell subsets are affected by Gfi1b loss.

Figure 1f and 8c: include scale bar for each panel. For Figure 1f, it appears that images of WT cells are at a higher magnification than images of mutant cells; if this is the case, the authors should adjust the magnification so that they are the same. Also in Figure 1f, large image on left: what are all the small blue dots surrounding the main cell?

We verified that the images have been taken with the same magnification. The scale bars were added. The small blue dots are non-MK cells (other haematopoietic precursors) often present in the MK preparation by sedimentation on BSA.

Figure 2b: the alignment of the WRD sequence is confusing. Are these all sequences of Gfi1b from different species? Are the amino acid positions 106 to 111 identical for Gfi1b from all shown species? Is the muWRD sequence from mouse Gfi1b? How is the consensus sequence at the top derived? It appears that FSWD is completely conserved yet the consensus lists it as FS/TXX. Please clarify.

Figure 2b is now Figure 2g in the revised MS.

Yes, these are Gfi1b WRD sequences from different species listed in the left part.

The amino acid positions are those from human GFI1B. We have now clarified this in the Figure legend.

The muWRD is a mutated form of human GFI1B that we made for the IP experiments. The consensus was characterised in a previous study (the Reference was added to the Figure legend)

We clarified the 5 amino acids consensus in the revised figure as FSXXXL in which only the first two amino acids (FS) and the last one (L) are fixed (according to the ref 56).

Figure 3c-e: are these various mutant versions of Gfi1b expressed at equal levels? An immunoblot showing that each protein is present should be included. Are these transgenes Flag-tagged as indicated in Figure 2b?

They are expressed at equal levels and we added Figure 3l to address this point.

Figure 2b is now Figure 2g in the revised MS.

The transgenes indicated in Figure 2g are flag-tagged indeed.

Page 14: the following statement is poorly supported and lacks quantitation: "This finding indicates that loss of Gfi1b can lead to both up- and down-regulation of b-catenin target genes in primary hematopoietic cells and changes the balance between canonical and non-canonical Wnt target genes." To what extent is the balance between canonical and non-canonical Wnt signaling altered in Gfi1b deficient cells?

Our previous GSEA analysis (Figure 5c) on RNA-seq data shows deregulation of canonical Wnt target genes in HSCs and MK. We added a new RNA-seq data set performed with two sets of Gfi1b WT/KO MKs. Unsupervised clustering of the expression data using two gene sets of canonical and non-canonical Wnt signalling also indicated that the balance between canonical and non-canonical Wnt genes is altered in these cells when Gfi1b is absent (Figure 6e).

Figure 5b and 6c: references to each gene set should be provided.

Figure 6c has been removed from the MS

We provide the references in supplementary Table 1 and we refer to it in the Figure legend accordingly.

Figure 7b: this looks like RNA-seq data, not ChIP-seq as indicated in the figure legend.

This Figure was moved to the supplementary data file (now as supplementary Figure 2 in the

revised MS) and the legend was corrected accordingly

Throughout the figures the authors switch between “Gfi1b” (indicating mouse origin) and “GFI1B” (indicating human origin). Is this an intentional distinction? Please clarify.

“Gfi1b” was used when murine cells or mice were used and “GFI1B” when human cells were analysed. We have harmonized this throughout the MS.

Abstract: The abstract would benefit from some editing to streamline and clarify what the authors are trying to present. As it reads now, it is rather difficult to follow.

We have revised the abstract and hope it is clearer now.

Figure 2a: Table has a typo: Histone.

This was corrected

Figure 3c-h: Why have the authors now switched to LiCl as a Wnt stimulus? LiCl targets GSK3 and is woefully nonspecific. These assays should be performed in the context of Wnt stimulation, to demonstrate that this effect is due to Wnt. Also, much better GSK3 inhibitors are available and could be employed.

We agree with the reviewer. However, TOP/FOP assays with LiCl have been used by others before and were published: Monteagudo S et al, Nat Commun 2017 Fig 2c. and/or Cai J et al, Nat Commun 2017 Fig 5b. In Figure 3a we obtained very similar results also with Wnt3A conditioned media or a Wnt3A expression vector (Figure 3b). Following the suggestion of the reviewer, we have now repeated this reporter assay with a more specific inhibitor, CHIR99021, and obtained again similar results.

Figure 3i-j: All of the other data in this figure is reported as TOP/FOP. Why have the authors now eliminated the FOP normalization? These experiments should be repeated in the context of the FOP reporter with the Gfi1b binding sites for normalization, if we are to compare these assays.

Figure 3i-j are now Figure 3j-k in the revised MS.

We have modified the classical TOP luciferase vector and added two Gfi1b binding sites. For this new vector, a corresponding FOP control is not available. If we would insert a Gfi1b binding site into the FOP vector as well, we would simply see repression as with all previously tested reporter genes that contain Gfi1b binding sites. We have therefore chosen to show the results with the TOP vector containing TCF and Gfi1b sites only as RLU reads.

Reviewer #3 (Remarks to the Author):

The manuscript by Shooshtarizadeh et al, identifies physical association of b-catenin and other Wnt pathway components with the transcription factor Gfi1b by immuno-precipitation followed by mass spectrometry. The authors then endeavor to demonstrate Gfi1b mediated activation of Wnt/b-catenin signaling in HSC and megakaryocyte cellularity and function. They do so by demonstrating the impact of Gfi1b and its co-factor LSD1 on Wnt/b-catenin transcriptional regulation by promoter driven reporter assays and by Gfi1b/LSD1 dependent modulation of Wnt target gene expression in hematopoietic and non-hematopoietic cells. Finally, they assert that attenuation of Wnt signaling in the Gfi1b deficient HSCs and MKs, can be partially rescued by external treatment with Wnt3a.

Although the Gfi1b-Wnt pathway interaction is an interesting and novel finding and the authors perform several and diverse kinds of follow up experiments to address their combinatorial role in hematopoietic development, the paper does not lead to a mechanistically compelling scenario of the

co-operativity between these pathways in this complex and vital process. As discussed below much of the data is circumstantial and a number of unexpected and conflicting observations are not adequately addressed or resolved.

Figure 1: Does deletion of exons 2-4 produce a true null phenotype? Does a truncated protein result from the mutated transcript and if so does it have any residual or dominant negative function? This question is addressed in our previous paper (Ref 9), where we show absence of a Gfi1b protein in Mx-Cre Gfi1b^{fl/fl} cells. No Gfi1b protein was detected in Gfi1b null cells.

Figure 2: Extent of Gfi1b and LSD1 precipitation is not indicated in the mass spec data. Reasons for using 293T instead of hematopoietic cells not justified.

2c. b-catenin immunoprecipitates with P2AGfi1b and Gfi1b^{muWRD} ie region of Gfi1b interacting with b-catenin is undefined. 2d. Co-IP of b-catenin in K562 cells only marginally better with Gfi1b antibody relative to IgG control. 2e. HEL-Same for pontin52; b-catenin not shown. 2f-g. Residual LSD1 precipitated with b-catenin in SNAG mutants not explained. Is the association of LSD1 with b-catenin partly Gfi1b independent?

Figure 2c is now Figure 2h in the revised MS.

Figure 2d-g is now Figure 2j-m in the revised MS.

293 cells were used for the ease of transfection and IP. We now show spectral counts and % coverage (Figure 2a, c). Also: to confirm our data, we performed a new Gfi1b-BIRA BioID experiment (Figure 2d-f and Figure 2o, p), which also resulted in the enrichment of proteins involved in the Wnt regulatory pathway. In addition, we have now completed a new Ch-IP seq experiment set in hematopoietic cells (K652), which demonstrates that GFI1B and b-catenin co-occupy sites at promoters and enhancers. Together with co-immune precipitations data and additional (new) BioID experiment with b-catenin, we believe that the interaction between GFI1B and b-catenin is well supported. Hence, the initial experiment in HEK 293 cells was suitable to give us the indication for new GFI1B interaction partners.

We have made new GFI1B mutants and were able to identify the C2H2 zinc finger domain of GFI1B to be involved in the interaction with b-catenin (see new Figure 2i). LSD1 was not detected in the GFI1B-Flag IP followed by mass spec (Fig. 2a), but was found in the GFI1B-BirA BioID experiment (Fig2d-f).

We repeated the Co-IP of b-catenin in K562 and HEL cells using all commercially available antibodies against Pontin52, Gfi1b and b-catenin. We also tried to reduce the amount of background by doing more washes. Nevertheless we were not able to obtain a cleaner co-IP. But we still believe that there is an enrichment of these proteins over background and we would like to keep these data rather than excluding them from the manuscript, unless specifically requested. Of note, we performed our second BioID experiment using b-catenin-BIRA and found again an association of Gfi1b and b-catenin, which supports our IP data (Figure 2o, p) and the notion that GFI1B and b-catenin interact. Of note, b-catenin is not highly expressed in the HEL cell line.

It was shown that b-catenin can bind independently to LSD1 in several previous and independent studies such as Song 2015 (Mol Cancer Res. Jun; 13(6): 969–981. Figure 6A) and Yakulov 2013 (Mol Cell Proteomics, Jul; 12(7): 1980–1994. Figure 4A). It is possible that the association of LSD1 with b-catenin is partly Gfi1b independent as indicated by Figure 2l, however the majority of b-catenin interaction with Gfi1b at target regions seems to be in the presence of LSD1 (Figure 7a and 7g).

Figure 3: Why is transcriptional activation shown as a ratio of TOP/FOP luciferase activity and not relative to the TOP vector alone (except in i and j) is unclear.

3c-e. Why are different WRD mutants used in reporter assays relative to those in the IP experiments? Also if the WRD mutant continues to bind b-catenin, TLE1/3 and CtBP and is as effective as wt Gfi1b in stimulating TCF mediated transcription, then what is its significance? What Gfi1bmutGBS stands for is not explained in the text or figure legend. 3f. Effect of P2AGfi1b/LSD1 combination on TCF reporter activity would also be more informative in demonstrating the physical basis of Gfi1b/LSD1 transcriptional synergy.

Figure 3f is now Figure 3g in the revised MS.

TCF dependent transcription is generally measured as a ratio of TOP/FOP to normalize for background promoter binding. In the case of Fig. 3i and j (now Fig. 3j and k), we have modified the classical TOP luciferase vector and added two Gfi1b binding sites. For this new vector, a corresponding FOP control is not available. If we would insert a Gfi1b binding site into the FOP vector as well, we would simply see repression as with all previously tested reporter genes that contain Gfi1b binding sites. Thus, since TOP is modified with the addition of Gfi1b motifs, adding them to FOP may cancel out a detectable impact. We have therefore chosen to show the results with the TOP vector containing TCF and Gfi1b sites only as RLU reads.

We mutated different residues of the consensus sequence in order to identify the most important amino acid. WRD mutants were not as effective as wt Gfi1b (the differences compared to wt Gfi1b were statistically significant (see Figure 3c-e).

We corrected "Gfi1bmutGBS" to Gfi1b-muWRD. We apologize for this mistake.

The P2AGfi1b/LSD1 combination was still as efficient as wtGfi1b/LSD1 on TCF reporter activity (data not shown) due to (in part) overexpression of LSD1 as opposed to endogenous level.

Figure 4c. Shift in intensity of lacZ expression (as a measure of TCF/LEF driven transcription) in gfi1b deficient versus wt cells would be more obvious if their profiles were superimposed rather than of being shown side by side relative to axin+/+ profiles which are lacZ negative.

How many mice were used for these analyses and statistical significance are not addressed?

No positive or negative controls are shown to establish the specificity versus randomness of the results.

The reason we superposed them with Gfi1b WT or KO mice (Axin +/+) was to subtract the endogenous beta galactosidase activity.

Five pairs of mice were used (each line represents one pair of Gfi1b WT/KO mice). We modified the text as follows to address the statistical significance of Gfi1b KO MKs: "Quantification of mean fluorescence intensities (MFI) of LacZ in these cells showed that the effect of Gfi1b deficiency on the Axin2 promoter activity was measurable in all three cell subsets and was statistically significant in MKs (Fig. 4c)."

This is a well-established mouse model to study Wnt signalling. We also performed a lacZ staining of LSKs and HSCs following treatment with LiCl and Wnt3a that shows an increase in lacZ MFI (supplementary Figure 1d)

Figure 5: Reasons or mechanisms for up- versus down-regulation of Wnt target genes were not adequately addressed upon Gfi1b deletion given that the latter is a putative positive regulator of b-catenin transcription.

We have addressed this issue and now discuss the possibility that a tripartite Gfi1b/b-catenin/LSD1 complex can act as an activator or as a repressor depending on cellular context. Examples of

complexes that contain LSD1 and act in both capacities (repressor or activator) have recently been published) see paper from He et al., *Oncogene* (2018) 37, 534–543.

Figure 6: This figure is rather artifactual and difficult to interpret as it is a compilation of data from different sources and cells (HPC7 versus ES). In order to determine common targets of Gfi1b and b-catenin in HSCs or megakaryocytes, ChIP experiments need to be performed in parallel for both proteins in the same cells under identical conditions. As such figure 6A shows only minute overlap between Gfi1b and b-catenin genomic targets. ~10% of Gfi1b targets are common to b-catenin and <4% of b-catenin targets are bound by Gfi1b. Is this extent of overlap statistically or biologically meaningful or even specific.

Figure 6 is now supplementary Figure 3a and supplementary Figure 4 a-c in the revised MS. As suggested by the reviewer we tried and succeeded to do ChIP-seq experiments for Gfi1b, b-catenin and LSD1 (all at endogenous level) under the same conditions and at the same time in K562 (a relevant haematopoietic cell line) with and without Wnt3A treatment. Of note, this will be the first available ChIP-seq data set for endogenous b-catenin at endogenous expression levels and as an untagged version. This is also a likely explanation why there are less peaks called compared to previously published b-catenin ChIP-seq data using knock in tagged b-catenin in ES cells (now supplementary Figure 3a and supplementary Figure 4 a-c). With our new ChIP-seq experiments, we found 93% of b-catenin peaks overlap with Gfi1b peaks at promoters before Wnt3A treatment. Once treated with Wnt3A there are still 82% of b-catenin peaks that overlap with Gfi1b at promoters (Figure 7a and supplementary Figure 6).

Interestingly, our results show that these overlaps between Gfi1b and b-catenin peaks happen with higher preference at promoters than elsewhere in the genome confirming our previous observations using previously published ChIP-seq data from different cell lines. Therefore we decided to keep the information of the old Figure 6 but move it to supplementary Figures in the revised version of the manuscript (it is now supplementary Figure 3a and supplementary Figure 4 a-c).

Figure 7: Unclear what proteins are being ChIPed in (a) and (b) or what cells were used in (a). ChIP results in 293 cells over-expressing b-catenin or Gfi1b may produce artifactual results. Why was enrichment of endogenous proteins at these loci in hematopoietic cells not determined instead? Unclear why acetylated H3K9 levels are being monitored in (g) instead of H3K4/K9 methylation levels.

Figure 7a showed representative ChIP-seq tracks based on previously published ChIP-seq data in HPC-7 and ES cells. We moved these data to supplementary material as supplementary Figure 3b and we clarified the Figure legend accordingly. Figure 7b (now supplementary Figure 2) are representative RNA-seq tracks obtained with sorted cells as mentioned and explained in the Figure.

Figure 8: Partial “rescue” of Gfi1b ko phenotype by b-catenin overexpression does not necessarily imply a common pathway but may be due to compensation from distinct and parallel pathways. Since the mechanism by which Gfi1b and LSD1 mediate transcriptional repression are well established, their anomalous behavior in “activating” b-catenin mediated transcription needs to be more substantially addressed. Particularly the following -

1. Although these actions are proposed to be independent of the DNA binding activity of Gfi1b, the authors do not directly demonstrate recruitment of Gfi1b/LSD1 to Wnt targets by b-catenin or related factors ie in b-catenin deficient or depleted cells.
2. The authors also do not attempt to explain the mechanism of Gfi1b mediated enhancement versus reduction of b-catenin enrichment on subsets of target promoters.
3. Since the SNAG domain of Gfi1b recruits LSD1 it is unclear why is the latter absent on certain b-catenin/Gfi1b targets and present on others.

4. Does the presence or absence of LSD1 impact the chromatin status of the loci particularly H3K4 and H3K9 methylation since these two residues are known to be demethylated by LSD1.
5. Since Gfi1b also interacts with HDACs (previously shown and in Fig 2), does its presence on Wnt targets lead to HDAC recruitment.

1- While we have not done experiments with b-catenin depleted cells, our new motif enrichment analysis shows that regions co-occupied by b-catenin and Gfi1b don't show enrichment of GFI1B consensus DNA binding motifs suggesting the presence of GFI1B at these sites to be independent of its ability to bind DNA (new Figure 7c). The analysis also shows an enrichment of TCF sites at region occupied by the GFI1B/b-catenin/LSD1 complex. This would also be in agreement with data from Fig. 2k, which indicates that Gfi1b can regulate TCF dependent transcription without directly binding to DNA.

2- This comment concerns more the old Figure 7c (ChIP b-catenin-flag PCR in 293 cells), which we removed from the revised manuscript. The exact mechanisms of transcriptional regulation of Wnt/b-catenin target genes by the GFI1B/b-catenin/LSD tripartite complex need to be clarified by future studies. At this point we have shown that such a complex exists at genomic sites and that the regulation of Wnt/b-catenin target genes depends on the status of Gfi1b. Whether the expression of target genes occupied by this complex will increase or decrease is very likely context dependent.

3- Our new Chip-Seq data indicate that several situations exist (see supplementary Figure 6), but clearly the majority of b-catenin peaks also contain both GFI1B and LSD1 at promoters (82% before Wnt treatment and 63% after treatment see Figure 7a). It is possible that other factors can take the place of LSD1 in a GFI1B/b-catenin complex.

4- We found that the presence of one or more factors (GFI1B/LSD1/b-catenin) led to a majority of target genes having an increase in H3K4me1 and a decrease in H3K9me2 (see supplementary Figure 6c). Although this is outside of the scope of this study, it would be of interest to investigate the impact of LSD1 on histone modification in the context of GFI1 or GFI1B, in particular given recent results by the Somerville group indicating that LSD1 associates with GFI1 (a close paralogue of Gfi1b) via its enzymatic site but rather acts in a structural capacity rather than as a histone demethylase at target sites (Cell Rep. 2018 Mar 27;22(13):3641-3659). It is possible that a similar situation is seen here with GFI1B.

5- We were not able to include anti H3K9 antibodies in our new Ch-IP seq experiment and thus cannot answer this question. We have to refer this to a follow-up study of our work.

Additionally other questions that need to be addressed include-

1. Do Wnt signaling or b-catenin association impact DNA binding dependent transcriptional effects of Gfi1b?
2. A coherent model that explains the data and ties in Wnt signaling with Gfi1b/LSD1 in HSC and megakaryocyte development and differentiation.

1- Since Gfi1b regulates its own promoter, we used it as an example of a Gfi1b target gene in reporter assays to verify if up-regulation of Wnt signaling following CHIR99021 (a GSK3beta inhibitor) treatment has an impact on Gfi1b's repressor activity on its own promoter (supplementary Figure 1c, bottom). Our result shows that this is not the case and that Gfi1b's activity on its target genes remains unaltered.

2- We have included a hypothetical model that integrates our findings and would explain the action of Gfi1b in the context of Wnt signalling in HSCs and MKs in Figure 8.

Reviewers' comments:

Reviewer #1 (Remarks to the Author):

- The CHIPseq analysis in the hematopoietic cell line K562 are very convincing and a good compromise for not being able to perform these type of analysis in HSCs or MKs but still having performed them now in a hematopoietic cell background
- The different outcome of deletion of Gfi1b for HSCs and MKs is clear and nicely shown in previous publications. However, the difference in impact in the 2 different cell populations would suggest a difference in mechanism downstream of Gfi1b deletion. The data presented in this manuscript do not provide an answer here yet

Reviewer #2 (Remarks to the Author):

The authors have adequately addressed the bulk of my initial comments and critiques. I do not have any further critiques. There are only some minor issues that need to be addressed: Some figures are unnecessarily busy and could be simplified. For example, the gating strategy shown in Fig. 1b and the mass spec data in Fig. 2c could be moved to Supplemental data. Figure 2b does not add much value: indicating the amount of enrichment or the p-value for "positive regulation of Wnt signaling pathway" would suffice. Proximity ligation using BioID is a well-established methodology and the diagram in Fig. 2d is not needed.

Why are the LSD1 blots in panels h and i of Fig. 2 qualitatively so different?

What is the added value of Fig. 3d? Are the first 3 bars the same as shown in panel a? And what are the L111A and F106A mutations in GFI1B? These are never mentioned in the text.

Fig. 3a-i: I assume all y-axes are TOP/FOP, however some plots lack labeling on the y-axis.

Fig. 6e: these heat maps are confusing; are they perhaps mislabeled? Specifically, the heat map on left appears to be flipped: the light blue KO columns are labeled WT1 and WT2 and the dark blue WT samples are labeled KO1 and KO2.

Fig. 7d-f and Suppl. 4a-b: this display of the data is difficult to understand. Can the authors use a different schematic to illustrate their point, e.g. pie charts as shown in Fig. 7h?

Reviewer #3 (Remarks to the Author):

The manuscript entitled "Gfi1b regulates the level of Wnt/ β -catenin signaling in hematopoietic stem cells and Megakaryocytes" by Shooshtarizadeh et.al., is a revised submission of a similarly titled manuscript which the authors had submitted previously. As such the authors have made a concerted even valiant effort to substantively address the numerous and disparate issues raised by the reviewers. Yet despite all their efforts the authors have not put forth a compelling mechanistic scenario for the collaborative engagement of β -catenin and Gfi1b/LSD1 in HSC and megakaryocyte maintenance and function and for the apparently novel transcriptional activity of Gfi1b documented in this study. Detailed below are some of the more prominent examples of the disconnect between their claims and results.

1. The co-IP of endogenous β -catenin with Gfi1b is very weak and barely above background noise despite strong β -catenin expression in K562 cells. Moreover this interaction was not documented in HEL cells (Fig2 j and k). Does this mean that the robust interaction between these proteins observed

upon over-expression in HEK293 cells is largely an artifact of the process?

2. The point mutation N290S abrogates DNA binding due to the disruption in the secondary structure of the zinc fingers. This eliminates the transcriptional repression ability of Gfi1bN290S relative to Gfi1b (Fig 3j). Gfi1b also associates with β -catenin via its zinc fingers and the results in Fig 3k demonstrate that Gfi1bN290S is virtually as active as wt Gfi1b in activating a TCF driven reporter presumably because it can still bind β -catenin. However the assumption underlying this crucial difference was not addressed. It is important to clarify this as the SNAG domain point mutant Gfi1bP2A is also transcriptionally inactive due to its inability to bind LSD1.

3. In page 17 they state that "reduction of GFI1B levels decreased the expression of CCND1 and BIRC5 (canonical Wnt genes) and increased the expression of NFAT5, ROCK2 and PTK7 genes (non-canonical Wnt genes) (Fig.5f)" and that these changes "correlated with lower or higher H3K9 acetylation at their promoters, respectively (Fig. 5g)". However the mechanisms responsible for these apparently opposing effects of Gfi1b at these promoters (if indeed they were Gfi1b dependent) were not addressed. Also how are other putative Wnt targets analyzed in subsequent figures impacted by Gfi1b levels?

4. In Fig. 7b the authors demonstrate binding of GFI1B and LSD1 to Axin2 and Yaf2 promoters both in the absence and presence of β -catenin and state the same in the 1st paragraph of page 19.

However in the very next paragraph the authors claim that "sites bound by GFI1B were significantly enriched for the GFI1B binding motif as well as the GATA binding motif genome-wide (Fig. 7c). However, sequences at sites co-occupied by β -catenin and GFI1B were found to be prominently enriched for the LEF1 motif as well as GATA factor motifs, but not for GFI1B's own binding motif (Fig.7c). This suggests that GFI1B does not necessarily have to directly contact DNA when in a complex with β -catenin. Of note, Wnt stimulation did not alter the majority of GFI1B and LSD1 targeted promoters and β -catenin bound most frequently to promoters where GFI1B and LSD1 are already present prior treatment (Fig 7d & Supplementary Fig. 6a)". This clearly represents a contradiction in terms and a major conceptual stumbling block of the study due to the following reasons.

(i) How do Gfi1b and LSD1 bind to the Axin1, Yaf2 and other Wnt responsive promoters in the absence of β -catenin? Since the sequences corresponding to binding elements are not shown it is unclear if they are at canonical Gfi or β -catenin/TCF binding sites.

(ii) Moreover the authors do not provide any evidence to show if recruitment of β -catenin to these promoters is dependent on Gfi1b/LSD1. Is the recruitment of β -catenin to these promoters impacted by absence or depletion of Gfi1b/LSD1?

(iii) Although the methylation state (H3K4me1 versus H3K9me2 levels) of these promoters (Fig 7b) is consistent with the activation state of the promoters, it is incompatible with the presence of LSD1 at these sites. Determining the impact of LSD1 depletion on the methylation state of these loci will clarify the mechanism.

5. Finally, although a partial rescue of the Gfi1b knock out phenotype by boosting Wnt signaling or over-expressing β -catenin demonstrates the potency of this pathway in HSC and megakaryocyte maintenance and function, this observation in fact undermines the proposed mechanism by which these factors collaborate to ensure the proper development and functions of these cells. If the facilitation of β -catenin recruitment by Gfi1b and LSD1 to common genomic targets is a significant process in these cells then how can β -catenin carry out its functions in the absence of Gfi1b and even

partially rescue the *gfi1b* ko phenotype? It may be that β -catenin is epistatic to *Gfi1b*, however the mechanism the authors propose in the paper require their collaborative action.

Overall the authors claim that *Gfi1b*/*LSD1* interact with both canonical and non-canonical Wnt signaling components and stimulate both pathways which often produce opposing effects. However the results as presented neither provide a plausible mechanistic explanation for these opposing effects nor for how *Gfi1b* discriminates between these alternative pathways.

Reviewers' comments:

Reviewer #1 (Remarks to the Author):

- The CHIPseq analysis in the hematopoietic cell line K562 are very convincing and a good compromise for not being able to perform these type of analysis in HSCs or MKs but still having performed them now in a hematopoietic cell background

- The different outcome of deletion of Gfi1b for HSCs and MKs is clear and nicely shown in previous publications. However, the difference in impact in the 2 different cell populations would suggest a difference in mechanism downstream of Gfi1b deletion. The data presented in this manuscript do not provide an answer here yet

We thank the reviewer for their feedback. Indeed, Gfi1b deletion has several effects such as the expansion of HSCs and MKs, which is reversible by Wnt treatment and the decrease in Axin2 reporter activity in mice (Fig. 4b and c). However, there are clear differences in the expression of Wnt target genes in HSCs and MKs when Gfi1b is deleted (see Fig. 5d, compare HSCs and MKs). This may be due to the differential expression of the key transcription factors and other gene regulators in HSCs and MKs. For example, self-replicating and pluripotent HSCs express Gfi1 and PU.1, while the more differentiated MKs express Gata1 and Fli1 which are not prominent in HSCs. These transcription factors and possibly other gene transcription regulators most certainly influence expression of Gfi1b target regions in HSCs and MKs and subsequently also lead to differences in the expression of Wnt target genes. The details of these differential regulatory networks will have to be unveiled by further studies.

Reviewer #2 (Remarks to the Author):

The authors have adequately addressed the bulk of my initial comments and critiques. I do not have any further critiques. There are only some minor issues that need to be addressed:

Some figures are unnecessarily busy and could be simplified. For example, the gating strategy shown in Fig. 1b and the mass spec data in Fig. 2c could be moved to Supplemental data.

Figs 1b and 2c were moved to supplementary Figures. Fig. 1b is now supplementary Fig. 1b and Fig. 2c is now supplementary Fig. 1c.

Figure 2b does not add much value: indicating the amount of enrichment or the p-value for "positive regulation of Wnt signaling pathway" would suffice.

Fig. 2b (now supplementary Fig. 1d) was moved to the supplementary figures.

Proximity ligation using BioID is a well-established methodology and the diagram in Fig. 2d is not needed.

The diagram was removed.

Why are the LSD1 blots in panels h and I of Fig. 2 qualitatively so different?

Panel h is now called panel d. Panel I is now called panel e.

Experiments for the blots in panel d were done more recently following a reviewer request (indicate which request) with different anti LSD1 Abs (antibody product no. and company) than in panel e (antibody product no. and company). We re-cropped the blot to avoid confusion and to be coherent with other LSD1 blots in this Figure.

What is the added value of Fig. 3d? Are the first 3 bars the same as shown in panel a? And what are the L111A and F106A mutations in GFI1B? These are never mentioned in the text.

L111A and F106A are shown in Fig 2c. The first 3 bars are repeated for comparison with these point mutated forms of GFI1B. We did these point mutations to identify the most important amino acid residue for this activity. We have clarified this in the text of the revised version (page 7).

Fig. 3a-i: I assume all y-axes are TOP/FOP, however some plots lack labeling on the y-axis.

Yes, the same labels have now been put on the y-axis of all Figure parts.

Fig. 6e: these heat maps are confusing; are they perhaps mislabeled? Specifically, the heat map on left appears to be flipped: the light blue KO columns are labeled WT1 and WT2 and the dark blue WT samples are labeled KO1 and KO2.

We thank the reviewer for pointing out the mislabeling in Fig.6e. We have now corrected it.

Fig. 7d-f and Suppl. 4a-b: this display of the data is difficult to understand. Can the authors use a different schematic to illustrate their point, e.g. pie charts as shown in Fig. 7h?

Fig. 7d is now called supplementary Fig 7d.

Suppl 4a-b are now called supplementary Fig 5a-b)

We have tried Pie charts and feel that the distribution charts (GraphPad) has the advantage of providing an estimate of the percentages (with each dot representing 1% of the distribution) We would prefer to keep the distribution chart as is in the paper. However, if the reviewer prefers, we will be happy to provide Pie charts.

Reviewer #3 (Remarks to the Author):

The manuscript entitled “Gfi1b regulates the level of Wnt/ β -catenin signaling in hematopoietic stem cells and Megakaryocytes” by Shooshtarizadeh et.al., is a revised submission of a similarly titled manuscript which the authors had submitted previously. As such the authors have made a concerted even valiant effort to substantively address the numerous and disparate issues raised by the reviewers. Yet despite all their efforts the authors have not put forth a compelling mechanistic scenario for the collaborative engagement of β -catenin and Gfi1b/LSD1 in HSC and megakaryocyte maintenance and function and for the apparently novel transcriptional activity of Gfi1b documented in this study. Detailed below are some of the more prominent examples of the disconnect between their claims and results.

We recognize that our work has not fully elucidated the mechanism by which GFI1B affects the Wnt/ β -catenin signaling pathway. However, we believe that we provide several independent lines of strong experimental evidence for a functional link between GFI1B and β -catenin and for potential mechanisms. We have revised the manuscript in order to adjust our mechanistic claims and more clearly present the parts that are clearly established and those that are suggested but not yet fully supported.

The evidence for a link between GFI1B and β -catenin that we provide is: i) immune-precipitations and mass spectrometry, ii) two independent Bio-ID experiments with different baits (i.e. Gfi1b and β -catenin), iii) TCF dependent reporter assays, iv) experiments with reporter mice using two different models, v) Chip-Seq experiments from the literature showing co-occupation of Gfi1b and β -catenin, vi) new independently done Chip-Seq experiment in K562 cells by us before and after Wnt treatment also showing co-occupation of endogenous GFI1B and β -catenin with p-values indicating statistical significance, vii) motif analysis showing overlap of TCF and GFI1B binding sites to a subset of co-occupied genes, viii) RNA-Seq and GSE analyses showing deregulation of Wnt/ β -catenin target genes in Gfi1b deficient cells, ix) co-immune precipitation of endogenous GFI1B/ β -catenin complexes shown in western blots, x) reduction of GFI1B's activation of TCF dependent transcription by β -catenin inhibitors, xi) reduction of the expansion of Gfi1b deficient HSCs and MKs by Wnt3a, xii) Regulation of β -catenin occupation of certain genes by GFI1B knockdown using ChIP-qPCR in K562 cells expressing GFI1B sh-RNA, xiii) Identification of a domain in GFI1B responsible for the binding to β -catenin.

We propose that GFI1B binds to a complex with β -catenin and LSD1 to a set of target promoters and regulates β -catenin target genes – this regulation is context dependent and other factors may play a role that are yet to be identified. In addition, we propose also that GFI1B influences β -catenin signaling by sequestering negative regulators of the pathway away from β -catenin (such as TLE factors). Below, we further address the specific points of the reviewer and elaborate on the evidence for these mechanisms.

1. The co-IP of endogenous β -catenin with Gfi1b is very weak and barely above background noise despite strong β -catenin expression in K562 cells. Moreover this interaction was not documented in HEL cells (Fig2 j and k) . Does this mean that the robust interaction between these proteins observed upon over-expression in HEK293 cells is largely an artifact of the process?

Fig2 j and k are now called Fig 2f and g.

To address the reviewer's comment we have now repeated the co-IP experiment using K562 cells and can now show a clearer association between endogenous GFI1B and endogenous β -catenin upon CHIR99021 treatment (see Fig 2h). Furthermore, there are numerous other lines of evidence within our study, which support the existence of an

interaction between these proteins such as: i) the immune precipitation and mass spectrometry results, ii) the BioID experiments with GFI1B and with beta-catenin (Fig. 2l) and iii) the co-localization of GFI1B and beta-catenin by different ChIP-seq experiments (Fig.7). Moreover, we have shown that the zinc finger domain in GFI1B is necessary for this the interaction (see Figure 2e). Lastly, the TOP/FOP reporter assays provide, albeit indirect, evidence and support for an interaction between GFI1B and β -catenin. These represent several, entirely independent lines of evidence for this claim.

2. The point mutation N290S abrogates DNA binding due to the disruption in the secondary structure of the zinc fingers. This eliminates the transcriptional repression ability of Gfi1bN290S relative to Gfi1b (Fig 3j). Gfi1b also associates with β -catenin via its zinc fingers and the results in Fig 3k demonstrate that Gfi1bN290S is virtually as active as wt Gfi1b in activating a TCF driven reporter presumably because it can still bind β -catenin. However the assumption underlying this crucial difference was not addressed. It is important to clarify this as the SNAG domain point mutant Gfi1bP2A is also transcriptionally inactive due to its inability to bind LSD1.

We have addressed this now by a new immunoprecipitation experiment that clearly shows that GFI1B N290S still can associate with β -catenin (as opposed to the GFI1B Δ Zinc mutant, which lacks all zinc finger domains). We added this new result to the paper (see Fig 3l) and mentioned this in the text (page 8) as suggested by reviewer.

3. In page 17 they state that “reduction of GFI1B levels decreased the expression of CCND1 and BIRC5 (canonical Wnt genes) and increased the expression of NFAT5, ROCK2 and PTK7 genes (non-canonical Wnt genes) (Fig.5f)” and that these changes “correlated with lower or higher H3K9 acetylation at their promoters, respectively (Fig. 5g)”. However the mechanisms responsible for these apparently opposing effects of Gfi1b at these promoters (if indeed they were Gfi1b dependent) were not addressed. Also how are other putative Wnt targets analyzed in subsequent figures impacted by Gfi1b levels?

We highlighted some deregulated canonical Wnt genes on the heat map in Figure 6e. In addition, in a new experiment we now show the impact of GFI1B knockdown on canonical and non-canonical Wnt pathway genes (see Fig. 7d). ChIP-qPCR analysis of K562 stable clones expressing either GFI1B specific (S23) or scrambled (C5) shRNA confirms the knockdown of GFI1B (upper part) at genes normally targeted by both GFI1B and β -catenin and also shows that β -catenin occupation is dependent on GFI1B. For instance, *Ccr12*, *Yaf2* and *SLC38A8* show a decrease in β -catenin enrichment in GFI1B KD K562 cells, while β -catenin enrichment at *Cdh1*, *Axin2* and *Sp5* seem unaffected. The potential mechanism behind this difference remains the subject of future investigation.

4. In Fig. 7b the authors demonstrate binding of GFI1B and LSD1 to Axin2 and Yaf2 promoters both in the absence and presence of β -catenin and state the same in the 1st paragraph of page 19. However in the very next paragraph the authors claim that “sites bound by GFI1B were significantly enriched for the GFI1B binding motif as well as the GATA binding motif genome-wide (Fig. 7c). However, sequences at sites co-occupied by β -catenin and GFI1B were found to be prominently enriched for the LEF1 motif as well as GATA factor motifs, but not for GFI1B's own binding motif (Fig.7c). This suggests that GFI1B does not necessarily have to directly contact DNA when in a complex with β -catenin. Of note, Wnt stimulation did not alter the majority of GFI1B and LSD1 targeted promoters and β -catenin bound most frequently to promoters where GFI1B and LSD1 are already present prior treatment (Fig 7d & Supplementary Fig. 6a)”. This clearly represents a contradiction in terms and a major conceptual stumbling block of the study due to the following reasons.

(Fig 7d was moved to supplementary data as supplementary Fig 7d and Supplementary Fig. 6a is now Supplementary Fig. 7a)

(i) How do Gfi1b and LSD1 bind to the Axin1, Yaf2 and other Wnt responsive promoters in the absence of β -catenin? Since the sequences corresponding to binding elements are not shown it is unclear if they are at canonical Gfi or β -catenin/TCF binding sites.

Axin2 and Yaf2 are examples of promoters where GFI1B binds only after Wnt treatment, although LSD1 is indeed present prior to treatment (Fig. 7b). We have modified the text to make this clearer (See page 11). In the case of the Axin2 promoter, it contains a consensus binding motif for both GFI1B and β -catenin. We highlighted this motif on the Fig 7b with an arrowhead. In the case of Yaf2, there are no consensus sequences for either factor.

However, the reviewer is correct in stating that Wnt stimulation does not alter the majority of GFI1B and LSD1 binding and that β -catenin binds most frequently to promoters where GFI1B and LSD1 are already present prior to treatment, and the question is therefore pertinent. When we analyze all sites that have GFI1B binding prior to Wnt treatment and binding of both GFI1B and β -Catenin following treatment, we find that none of these sites contain a perfect consensus GFI1B motif.

As the reviewer notes, our results show that GFI1B and LSD1 are present at most β -catenin binding sites prior to Wnt treatment. The fact that GFI1B binding motifs are less frequent at such sites (compared to sites bound by GFI1B but not β -catenin) suggests that GFI1B binding at these sites differs from its activity at other sites.

In general, the presence of a transcription factor's binding motif is not strictly required for binding. It is possible for transcription factors to bind in the absence of their established motif, and to have their nucleotide sequence preference altered by DNA shape, genomic context, DNA modifications and coding and noncoding (genetic) variation as highlighted in this review (Inukai et al. 2017, Current Opinion in Genetics and Development), or, alternatively being through another factor that already sits on the DNA, so that a direct contact with DNA is not necessary.

It is possible that GFI1B is present at sites prior to the recruitment of β -catenin as a form of "priming", and that once β -catenin is recruited to these sites, β -catenin and GFI1B form a new complex that is responsible for the regulation of the gene. The fact that recruitment of β -catenin to several promoters is reduced in the context of a GFI1B knockdown or deficiency supports this. An additional potential mechanism is that GFI1B may be interacting with negative regulators of β -catenin and sequestering them away from β -catenin, thus preventing their negative activity. Notably, these two mechanisms are not mutually exclusive and it is likely that they are both in play to some extent.

(ii) Moreover the authors do not provide any evidence to show if recruitment of β -catenin to these promoters is dependent on Gfi1b/LSD1. Is the recruitment of β -catenin to these promoters impacted by absence or depletion of Gfi1b/LSD1?

We thank the reviewer for this comment and have now done a GFI1B and β -catenin ChIP-qPCR in K562 cell lines stably transfected with a scramble (C5) or a GFI1B KD specific (S23) Sh-RNA treated with a GSK-3 inhibitor, CHIR99021, shown in Fig 7d. Interestingly, we find that GFI1B knockdown leads to a decrease in β -catenin enrichment at specific target genes, such as *SLC38A8*, *Ccr12* and *Yaf2*, indicating that GFI1B is essential for β -catenin recruitment to these gene promoters. One further line of evidence suggesting the importance of GFI1B for β -catenin recruitment to DNA is the fact that almost the totality of β -catenin binding sites overlaps with GFI1B. We have added p-values to the Venn Diagrams in Fig 7a showing the considerable statistical significance of the overlap between GFI1B and β -catenin binding (2.98e-192 without Wnt treatment, 1.92e-99 with Wnt treatment).

(iii) Although the methylation state (H3K4me1 versus H3K9me2 levels) of these promoters (Fig 7b) is consistent with the activation state of the promoters, it is incompatible with the presence of LSD1 at these sites. Determining the impact of LSD1 depletion on the methylation state of these loci will clarify the mechanism.

We thank the reviewer for their input, but respectfully disagree that the presence of LSD1 at sites with H3K4me1 and H3K9me2 is incompatible. A recent study by Maiques-Diaz et al (2018, Enhancer Activation by pharmacologic displacement of LSD1 from Gfi1 induces differentiation in acute myeloid leukemia. Cell Reports 22, 3641-3659) nicely showed that LSD1 demethylase activity is inhibited in the presence of a GFI1 N-terminal SNAG-domain. This is due to the substrate-binding pockets of LSD1 being occupied by the N-terminal SNAG domain, which mimics the structure of histone H3 N-terminal tail (Baron et al. 2011, Molecular mimicry and ligand recognition in binding and catalysis by the histone demethylase LSD1-CoREST complex. Structure 19(2):212-20.) By extension, the SNAG-domain of GFI1B would similarly render LSD1 enzymatically inactive upon binding, as the substrate-binding site cannot simultaneously bind GFI1B's SNAG-domain and H3K4me1/2. Also, similar to our finding that the presence of LSD1 and GFI1B is concomitant with H3K4me1 and H3K9me2 in K562 cells (Figure 7b), Maiques-Diaz et al. also show the co-localization of LSD1 and GFI1 with H3K4 mono-, di- and tri-methylation and that there is no significant difference in enrichment at these loci upon LSD1 chemical inhibition.

Saleque et al Mol Cell 2007, the first paper that highlighted Gfi1 and Gfi1b's interaction with LSD1, indicated LSD1 loss derepresses Gfi1b targets in MEL cells and leads to an accumulation of H3K4 methylation, detected by using an Abcam antibody that targets both H3K4me2 and H3K4me3 (ab6000). It is possible that the H3K4 methylation enrichment detected upon LSD1 depletion is due to a histone lysine-specific methyltransferase, rather than loss of LSD1 as a histone demethylase and gene transcriptional repression by Gfi1 or Gfi1b is mediated by histone deacetylases (HDACs) that also associate with Gfi1 and Gfi1b (Duan et al. MCB 2005). This would require further investigation

5. Finally, although a partial rescue of the Gfi1b knock out phenotype by boosting Wnt signaling or over-expressing β -catenin demonstrates the potency of this pathway in HSC and megakaryocyte maintenance and function, this observation in fact undermines the proposed mechanism by which these factors collaborate to ensure the proper development and functions of these cells. If the facilitation of β -catenin recruitment by Gfi1b and LSD1 to common genomic targets is a significant process in these cells then how can β -catenin carry out its functions in the absence of Gfi1b and even partially rescue the gfi1b ko phenotype? It may be that β -catenin is epistatic to Gfi1b, however the mechanism the authors propose in the paper require their collaborative action.

ChIP-qPCR shows that at some target genes GF11B is essential for β -catenin recruitment and at other promoter regions β -catenin could still bind target regions in the absence of GF11B (see Fig. 7d). This could indicate that there are some important target regions, which could be influenced by Wnt/ β -catenin signaling even without Gfi1b presence and hence enable a partial MK phenotype rescue. We have modified the discussion to include these observations (page 14).

Additionally, this observation is consistent with the alternate mechanism we propose where GF11B activates the Wnt β -catenin pathway by sequestering negative regulators of β -catenin away from β -catenin. This is consistent with the interaction partners identified in the Mass-Spec experiments showing interaction with negative Wnt pathway regulators as well as the luciferase assays that show that DNA binding activity of GF11B is dispensable for its effect on β -Catenin activation.

Overall the authors claim that Gfi1b/LSD1 interact with both canonical and non-canonical Wnt signaling components and stimulate both pathways which often produce opposing effects. However the results as presented neither provide a plausible mechanistic explanation for these opposing effects nor for how Gfi1b discriminates between these alternative pathways.

We agree and have both softened and clarified our mechanistic claims; we do not have strong evidence for a precise mechanism how exactly GF11B/ β -catenin complexes regulate gene expression. But we have a large body of evidence for GF11B – β -catenin interaction, functional link between both and co-occupation of promoters and enhancers (see our response above: IP-Mass spec, two Bio ID experiments, several co-Immune-precipitations, several independent ChIP seq experiments and data sets) and we have several RNA-Seq experiments showing regulation of Wnt target gene expression by GF11B. In addition, a sequestration of β -catenin inhibitors (e.g, TLE proteins) by GF11B leading to activation of transcription of β -catenin target genes may also be at play. Evidence for this comes from our IP-mass spectrometry, Bio-ID results, co-immune precipitations and from reporter gene experiments. Reviewers 1 and 2 appear comfortable with the fact that these elements explain part of GF11B's action with regard to the Wnt/ β -catenin signalling pathway and we hope that this reviewer may also agree that we provide evidence for a first understanding how GF11B/ β -catenin complexes might work in gene regulation and that a more precise clarification and the full biochemical definition of this mechanisms will have to be the subject of future studies.

REVIEWERS' COMMENTS:

Reviewer #3 (Remarks to the Author):

The manuscript entitled "Gfi1b regulates the level of Wnt/ β -catenin signaling in hematopoietic stem cells and megakaryocytes" by Shooshtarizadeh et.al., is a substantially revised version of similarly titled ones submitted previously. In this version the authors have largely addressed most of the major concerns raised previously by all three reviewers. The additional results and corresponding explanations in conjunction with appropriate references to recent literature have considerably improved this study making it more suitable for publication in Nature Communications. The remaining issues that I have with this manuscript are as follows:

1. In Figure7a β -catenin binds to 247 promoters and 172 promoters in untreated versus Wnt3A treated K562 cells, respectively. This is an unexpected result that the authors do not address in the paper. The questions it raises include
 - a. What is the expression level of β -catenin in untreated versus Wnt3A treated cells?
 - b. Do the β -catenin targets in untreated versus treated cells match up?
2. The authors claim that recruitment of Gfi1b and LSD1 to TCF/LEF sites is independent of Gfi1b binding to DNA or at least to its consensus element. However they provide no evidence of what recruits Gfi1b/LSD1 to these promoters (as seen in Figure7b). Providing this answer would solidify their model considerably more than the hand waving explanation they provide in the Discussion that "It should be noted that consensus motifs are not strictly required for binding to DNA. It is possible for transcription factors to bind in the absence of their established motif, and to have their nucleotide sequence preference altered by DNA shape, genomic context, DNA modifications and coding and noncoding (genetic) variation".
3. Judging by numbers alone Gfi1b and LSD1 co-occupy and presumably regulate the majority of β -catenin targets (Figure7a). However the reverse is not true since β -catenin only co-occupies a very small fraction of Gfi1b/LSD1 targets (108 of 5727) in Wnt induced cells. This would suggest (though hardly prove) a more influential role of Gfi1b and LSD1 in Wnt/ β -catenin biology instead of the reverse. The ChIP results are also quite consistent with the previously shown IP results (Figure 2f versus 2i and j) with endogenous Gfi1b and β -catenin, respectively. However this scenario is not addressed in the paper.
4. The transplant experiments with gfi1b mutant cells into irradiated recipients (Suppl Fig 8) is not adequately explained. Since Gfi1b is essential for erythroid development (a fact that the authors tend to disregard for the most part) it is unclear what the contribution of transplanted mutant cells is to the total hematopoietic compartment of the irradiated recipients 4 months post transplant. Also β -catenin expression does not alter (rescue?) megakaryocyte numbers in animals transplanted with gfi1b deficient cells (Suppl Fig 8b).

Overall the authors report novel observations connecting Gfi1b/LSD1 to Wnt/ β -catenin biology and the likely ramifications of these interactions. However the data as presented is not entirely consistent with the proposed conclusions for reasons identified above. Although some of these issues could be resolved in subsequent reports it may be worthwhile to set the record straight here especially when proposing paradigm shifting mechanisms.

Answer to reviewer 3:

Reviewer #3 (Remarks to the Author):

The manuscript entitled “Gfi1b regulates the level of Wnt/ β -catenin signaling in hematopoietic stem cells and megakaryocytes” by Shooshtarizadeh et.al., is a substantially revised version of similarly titled ones submitted previously. In this version the authors have largely addressed most of the major concerns raised previously by all three reviewers. The additional results and corresponding explanations in conjunction with appropriate references to recent literature have considerably improved this study making it more suitable for publication in Nature Communications. The remaining issues that I have with this manuscript are as follows:

1. In Figure 7a β -catenin binds to 247 promoters and 172 promoters in untreated versus Wnt3A treated K562 cells, respectively. This is an unexpected result that the authors do not address in the paper. The questions it raises include

- a. What is the expression level of β -catenin in untreated versus Wnt3A treated cells?
- b. Do the β -catenin targets in untreated versus treated cells match up?

a. We thank the reviewer for this question. Indeed, we were also surprised to detect so many β -catenin peaks at promoters in the untreated K562 cells, especially given the significantly lower levels of β -catenin nuclear protein levels in untreated versus Wnt3A treated samples (supplementary Figure 6). One explanation could be that that baseline activity of canonical Wnt signalling in K562 cells (untreated cells) is in part due to β -catenin stabilization by BCR-ABL which is present in K562 cells (as shown in Chen Y et al, Leukemia 2010). Wnt3A treatment leads to activation of canonical Wnt pathway followed by increased stability of β -catenin protein and its nuclear translocation (as reviewed in the introduction section and shown in supplementary Figure 6). Although the number of detected β -catenin peaks at promoters was lower in Wnt3A treated cells (Figure 7a), we observed more β -catenin peaks at enhancers (Fig. 7 g-h) following Wnt3A treatment. Of note overall detected peaks in β -catenin Chip-seq were 603 in untreated and 815 in Wnt3A treated cells (supplementary Figure 7, right column).

b. The number of gene promoters bound by β -Catenin in both untreated and Wnt3A treated cells is 60. (supplementary Figure 7a, bottom Venn diagram). In the case of enhancers, there are 25 bound without Wnt3a and 92 bound with Wnt3a treatment. 14 of these are common to both situations. This suggests that the baseline activity of β -catenin in untreated cells is different from its activity following Wnt3a treatment. We have added text to the discussion addressing these observations (see revised MS on page 15).

2. The authors claim that recruitment of Gfi1b and LSD1 to TCF/LEF sites is independent of Gfi1b binding to DNA or at least to its consensus element. However, they provide no evidence of what recruits Gfi1b/LSD1 to these promoters (as seen in Figure 7b). Providing this answer would be solidify their model considerably more than the hand waving explanation they provide in the Discussion that “It should be noted that consensus motifs are not strictly required for binding to DNA. It is possible for transcription factors to bind in the absence of their established motif, and to have their nucleotide sequence preference altered by DNA shape, genomic context, DNA modifications and coding and noncoding (genetic) variation”.

To clarify, we hypothesize that GFI1B can be recruited in a complex with beta-catenin to sites where TCF binds DNA. In this situation GFI1B does not necessarily have to contact DNA directly and thus would not require a DNA binding motif to be present at this site. Independently of this, we also suggest an additional

potential mechanism whereby GFI1B could activate TCF-dependent transcription by sequestering negative regulators without binding DNA or occupying a specific site via another factor. This mechanism is also independent of DNA binding. At the moment, we have evidence supporting these models without excluding one or the other.

To clarify this in the text, we have added the following text to the discussion (page 16):

“Our data suggest that GFI1B recruitment to TCF/LEF sites can be independent of its consensus element. An alternative possibility is that GFI1B can be recruited in a complex with β -catenin to TCF at sites where TCF binds DNA. In this situation GFI1B does not necessarily have to contact DNA directly and thus would not require a DNA binding motif to be present at this site. Independently of this, we also suggest an additional potential mechanism whereby GFI1B could activate TCF-dependent transcription by sequestering negative regulators without binding DNA or occupying a specific site via another factor. This mechanism is also independent of DNA binding. At the moment, we have evidence supporting these models without excluding one or the other”.

3. Judging by numbers alone Gfi1b and LSD1 co-occupy and presumably regulate the majority of β -catenin targets (Figure 7a). However, the reverse is not true since β -catenin only co-occupies a very small fraction of Gfi1b/LSD1 targets (108 of 5727) in Wnt induced cells. This would suggest (though hardly prove) a more influential role of Gfi1b and LSD1 in Wnt/ β -catenin biology instead of the reverse. The ChIP results are also quite consistent with the previously shown IP results (Figure 2f versus 2i and j) with endogenous Gfi1b and β -catenin, respectively. However this scenario is not addressed in the paper.

We do agree with the reviewer's comment on “a more influential role of Gfi1b and LSD1 in Wnt/ β -catenin biology” as highlighted in the title of our paper “**Gfi1b regulates the level of Wnt/ β -catenin signaling in hematopoietic stem cells and megakaryocytes**”.

Indeed, our results do suggest that GFI1B and LSD1 are involved in multiple processes other than Wnt/ β -catenin whereas many β -catenin targets are likely affected by GFI1B and LSD1.

We added the following text to the discussion section (see revised MS on page 14) to address this scenario:

“Interestingly, the majority of β -catenin targets are co-occupied by GFI1B and LSD1, while β -catenin only co-occupies a very small fraction of Gfi1b/LSD1 targets (108 of 5727 in Wnt treated cells, see Fig. 7a). This may indicate that besides GFI1B and LSD1's influential role in Wnt/ β -catenin biology, both are involved in multiple other processes, whereas many β -catenin targets are affected by GFI1B and LSD1. In addition, it is notable that while the overall number of β -catenin peaks increases following Wnt3A treatment (from 603 to 815), there is a considerable number of promoters bound by β -catenin in the absence of Wnt3A (247), suggesting a certain baseline level of activity of β -catenin.”

4. The transplant experiments with gfi1b mutant cells into irradiated recipients (Suppl Fig 8) is not adequately explained. Since Gfi1b is essential for erythroid development (a fact that the authors tend to disregard for the most part) it is unclear what the contribution of transplanted mutant cells is to the total hematopoietic compartment of the irradiated recipients 4 months post transplant. Also β -catenin expression does not alter (rescue?) megakaryocyte numbers in animals transplanted with gfi1b deficient cells (Suppl Fig 8b).

We have modified the text and have added more explanations on this experiment (see revised MS on page 13).

HSC and MK expansion is the most pronounced feature of Gfi1b KO mice as presented in Figure 1 and in previous publications from us and other groups. Therefore, for the transplant experiments we focused on

these two compartments. We did not include erythroid development in this study, but we agree with the reviewer that indeed Gfi1b has a pertinent role in red cell differentiation. The role of β -catenin/Gfi1b interactions, if any, in this compartment will have to be the subject of another study.

The impact of β -catenin expression on megakaryocyte numbers in the transplant experiments was variable as opposed to HSCs numbers, which may be due to the experimental design or its limitations. We modified the text and added the following to address this issue:

“Flow cytometric analysis, four months post transplantation, showed that Gfi1b deficient β -catenin+ HSC frequencies were significantly lower than those of GFP controls (Supplementary Fig. 8a and b) suggesting that activating the canonical Wnt signaling pathway by retroviral expression of active β -catenin, inhibits in vivo expansion of Gfi1b KO HSCs.”

Overall the authors report novel observations connecting Gfi1b/LSD1 to Wnt/ β -catenin biology and the likely ramifications of these interactions. However the data as presented is not entirely consistent with the proposed conclusions for reasons identified above. Although some of these issues could be resolved in subsequent reports it may be worthwhile to set the record straight here especially when proposing paradigm shifting mechanisms.

We agree with the reviewer about the need to clarify which parts of our conclusions are fully supported by evidence and which are more speculative, especially at the level of the proposed mechanisms. We have made modifications to the text, as mentioned above, that we believe address this valid concern.